# On the pH-dependence of α-synuclein amyloid polymorphism and the role of secondary nucleation in seed-based amyloid propagation

Lukas Frey[1†], Dhiman Ghosh[1†], Bilal M Qureshi[2†], David Rhyner[1], Ricardo Guerrero-Ferreira[3], Aditya Pokharna[1], Witek Kwiatkowski[1], Tetiana Serdiuk[4], Paola Picotti[4], Roland Riek[1]*, Jason Greenwald[1]*

[1]Institute of Molecular Physical Science, Zürich, Switzerland; [2]Scientific Center for Optical and Electron Microscopy, Zürich, Switzerland; [3]Robert P. Apkarian Integrated Electron Microscopy Core, Emory University, Atlanta, United States; [4]Institute of Molecular Systems Biology, ETH Zürich, Zurich, Switzerland

*For correspondence:
roland.riek@phys.chem.ethz.ch
(RR);
gjason@ethz.ch (JG)

†These authors contributed equally to this work

Competing interest: The authors declare that no competing interests exist.

**Abstract** The aggregation of the protein α-synuclein is closely associated with several neurodegenerative disorders and as such the structures of the amyloid fibril aggregates have high scientific and medical significance. However, there are dozens of unique atomic-resolution structures of these aggregates, and such a highly polymorphic nature of the α-synuclein fibrils hampers efforts in disease-relevant in vitro studies on α-synuclein amyloid aggregation. In order to better understand the factors that affect polymorph selection, we studied the structures of α-synuclein fibrils in vitro as a function of pH and buffer using cryo-EM helical reconstruction. We find that in the physiological range of pH 5.8–7.4, a pH-dependent selection between Type 1, 2, and 3 polymorphs occurs. Our results indicate that even in the presence of seeds, the polymorph selection during aggregation is highly dependent on the buffer conditions, attributed to the non-polymorph-specific nature of secondary nucleation. We also uncovered two new polymorphs that occur at pH 7.0 in phosphate-buffered saline. The first is a monofilament Type 1 fibril that highly resembles the structure of the juvenile-onset synucleinopathy polymorph found in patient-derived material. The second is a new Type 5 polymorph that resembles a polymorph that has been recently reported in a study that used diseased tissues to seed aggregation. Taken together, our results highlight the shallow amyloid energy hypersurface that can be altered by subtle changes in the environment, including the pH which is shown to play a major role in polymorph selection and in many cases appears to be the determining factor in seeded aggregation. The results also suggest the possibility of producing disease-relevant structure in vitro.

## eLife assessment

This study presents **important** findings on the different polymorphs of alpha-synuclein filaments that form at various pH's during in vitro assembly reactions with purified recombinant protein. Of particular note is the discovery of two new polymorphs (1M and 5A) that form in PBS buffer at pH 7. The strength of the evidence presented is **convincing**. The work will be of interest to biochemists and biophysicists working on protein aggregation and amyloids.

## Introduction

Amyloid fibrils are a unique class of self-assembled filamentous protein/peptide aggregates that have been associated with dozens of human diseases including Alzheimer's, Parkinson's, type II diabetes, and prion diseases (*Chiti and Dobson, 2006*). It has long been established that all amyloids contain a common structural scaffold known as the cross β-sheet motif, yet only more recently, with the advent of solid-state nuclear magnetic resonance spectroscopy (NMR) (*Meier et al., 2017*) and cryo-electron microscopy (cryo-EM) helical reconstruction (*He and Scheres, 2017*), has it become clear that this shared repetitive motif can generate a wide range of structural folds (*Iadanza et al., 2018*; *Sawaya et al., 2021*). Yet unlike globular proteins, whose amino acid sequence typically guides the protein folding process into a unique 3D structure, the amyloid structures derived from a single peptide sequence are often polymorphic: they can exist in various stable 3D architectures. Another unique property of amyloids is their tendency to induce their own formation in a self-replicative manner. Amyloid formation is a nucleated growth process analogous to the growth of a 1D crystal (*Riek and Eisenberg, 2016*) and can therefore be kinetically enhanced through templating on preformed fibrillar structures, termed seeds. Thus, in seeding, the amyloid fold itself acts as a structural template for nucleation and growth, whereby protein monomers are induced into the amyloid conformation and contribute to fibril elongation. Additionally, as revealed by kinetic experiments, there are secondary nucleation processes, which include de novo nuclei formation on the lateral surface of the amyloid fibrils (*Törnquist et al., 2018*). Structural polymorphism and seeding-enhanced growth are two hallmarks of amyloid structural biology. However, despite their likely importance in the etiology of many diseases, the factors that control polymorphs and the molecular mechanisms of seeding are poorly understood.

Amyloid polymorphs whose differences lie in both their tertiary structure (the arrangement of the β-strands) and the quaternary structure (protofilament-protofilament assembly) have been found to display distinct biological activities (*Uemura et al., 2023*; *Peelaerts et al., 2015*; *Guo et al., 2013*; *Van der Perren et al., 2020*; *Bousset et al., 2013*). In the case of the protein tau, cryo-EM has been used to correlate 14 different tauopathies with distinct combinations of 8 different amyloid polymorphs (*Shi et al., 2021*). In the case of α-synuclein (α-Syn), the structures of many in vitro polymorphs have been determined (*Li et al., 2018*; *Li, 2018*; *Guerrero-Ferreira et al., 2018*; *Zhao et al., 2020b*; *Zhao et al., 2020a*; *Sun et al., 2020*; *Ni et al., 2019*; *Boyer et al., 2019*; *Guerrero-Ferreira et al., 2019*; *Boyer et al., 2020*; *Schweighauser et al., 2020*; *Long et al., 2021*; *Sun et al., 2021*; *Hojjatian et al., 2021*; *McGlinchey et al., 2021*; *Lövestam et al., 2021*; *Frieg et al., 2021*; *Sun et al., 2023*; *Zhao et al., 2023*; *Zhang et al., 2023*; *Yang et al., 2022*; *Frieg et al., 2022*; *Yang et al., 2023*; *Fan et al., 2023*) and yet, they are all distinct from the polymorphs that have been sarkosyl-extracted from the brains of patients of both multiple systems atrophy (MSA) (*Schweighauser et al., 2020*) and Parkinson's disease (PD) (*Yang et al., 2022*). Understanding how these disease-relevant polymorphs form and reproducing them in vitro will be critical in the study of these synucleinopathies. Some clues are present in the sarkosyl-insoluble brain-derived structures: two MSA polymorphs contain an unidentified molecule at the interface of the protofilaments and in the PD polymorph an unidentified molecule appears to be stabilizing the structure of the monofilament fibril. These findings support the idea that the polymorph type is in some way associated with the disease, and that the environment plays a critical role in amyloid formation, essentially dictating which polymorphs are accessible.

Concerning the seeding of polymorphs, it was demonstrated that prion strains with their specific pathologies can be serially transmitted from animal to animal (*Prusiner, 1998*; *Aguzzi and Rajendran, 2009*) and that strains can be amplified in vitro by protein misfolding cyclic amplification (PMCA) (*Saborio et al., 2001*; *Soto et al., 2002*) or similar approaches (*Srivastava et al., 2022*). However, in vitro, amyloid seeding is not always faithful in reproducing the seed polymorph. In the case of α-Syn, PMCA-based seeding has been proposed as a diagnostic tool that could also differentiate between MSA and PD (*Soto et al., 2002*; *Gerez and Riek, 2020*). However, attempts to amplify the MSA and the PD patient-derived polymorphs in vitro with recombinant α-Syn have so far resulted in structures distinct from those that were extracted with sarkosyl from patient brains (*Lövestam et al., 2021*; *Strohäker et al., 2019*; *Burger et al., 2021*). These results could be due to the strong role of secondary nucleation in the in vitro seeding processes, which in the case of α-Syn have been shown to involve flexible interactions that might not be able to transmit polymorph-specific structural information (*Gerez and Riek, 2020*). However, it could also be that multiple polymorphs exist in diseased

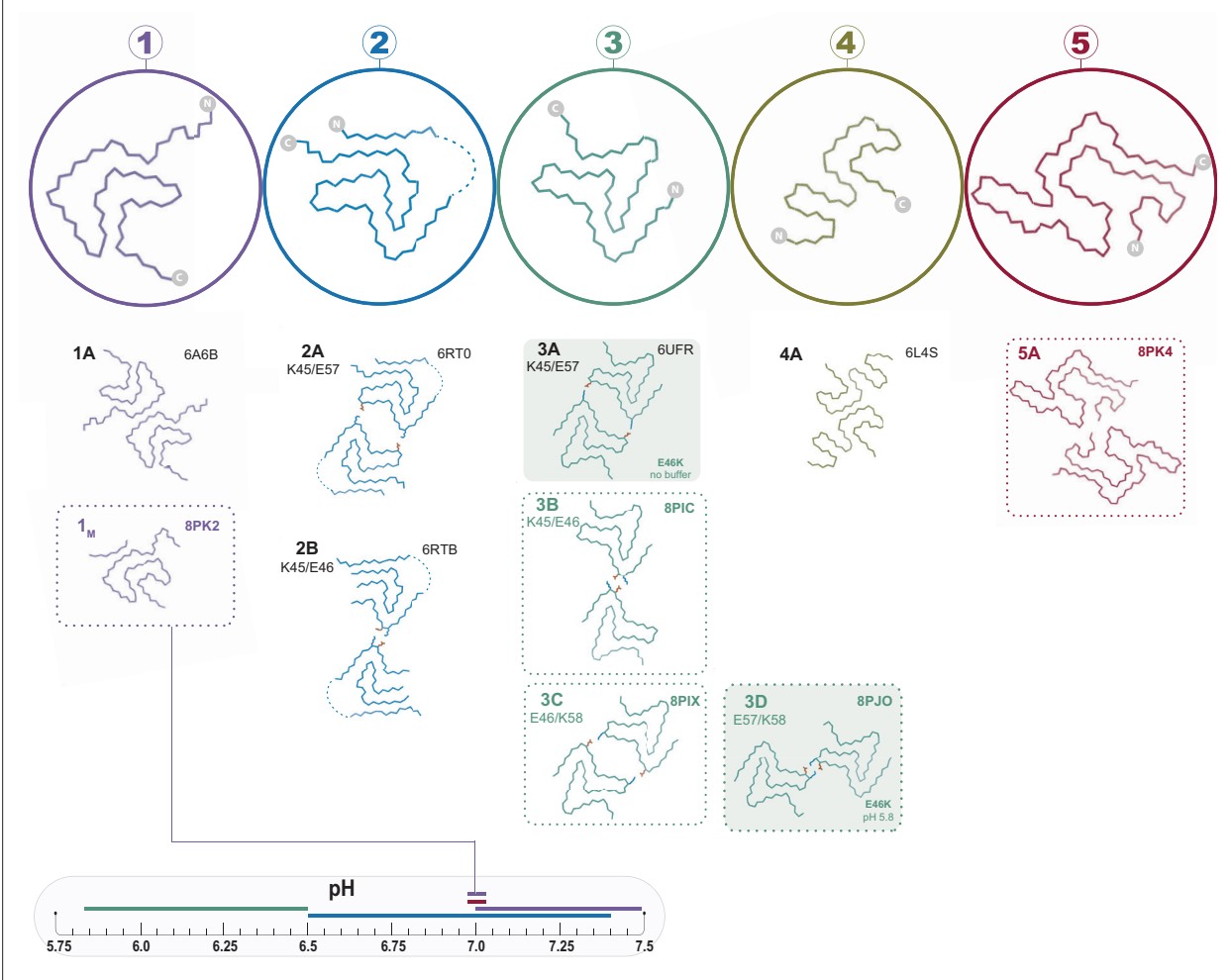

**Figure 1.** The polymorphs formed in vitro by wild-type α-synuclein (α-Syn). The numbered backbone Cα traces depicting a single chain from each of the five types of polymorphs that have been reported to form in vitro with wild-type α-Syn but without cofactors. The interface variants of each type are also shown by a representative structure from the PDB or from one of the structures reported in this manuscript (boxed in dotted lines). Two mutant structures for the Type 3 fold are included in order to show the special 3A and 3D interfaces that form for these mutants (shaded background). The observed pH-dependence of each of the polymorphs described in this manuscript is indicated by the colored line above the pH scale (or, in the case of the 1M polymorph reported here, by a line connecting it to the scale).

tissues and that the sarkosyl-based procedure itself selects different polymorphs compared to the PMCA-based procedures. Nonetheless, seeding with α-Syn disease-derived amyloids has, at least for cerebrospinal fluid (CSF)-derived seeds, some diagnostic utility as it been shown that under conditions of low α-Syn monomer, patient-derived CSF can seed several polymorphs while healthy CSF does not seed amyloid formation at all (*Fan et al., 2023*). On the other hand, seeding experiments with in vitro-derived polymorphs have yielded mixed results in that some polymorphs are seemingly easier to reproduce via seeding. For example, the Type 4 polymorph from E46K (*Long et al., 2021*) or G51D (*Sun et al., 2021*) mutants can be transmitted to wild-type α-Syn, and a polymorph that forms normally only at low salt can be produced at high salt and vice versa by using very high seed concentrations (10%) and by limiting primary and secondary nucleation processes (*Bousset et al., 2013*; *Peduzzo et al., 2020*).

Having observed the plethora of α-Syn fibril polymorphs published in just the last few years, we set out to investigate the relationship between α-Syn polymorphs and their aggregation conditions. The speed and abundance of polymorphs being discovered have led to inconsistent naming conventions, and in this work, we expand upon previous nomenclature (*Guerrero-Ferreira et al., 2018*; *Guerrero-Ferreira et al., 2019*) which is illustrated in *Figure 1*, whereby polymorphs with distinct folds are assigned different Arabic numerals and sub-polymorphs formed through different

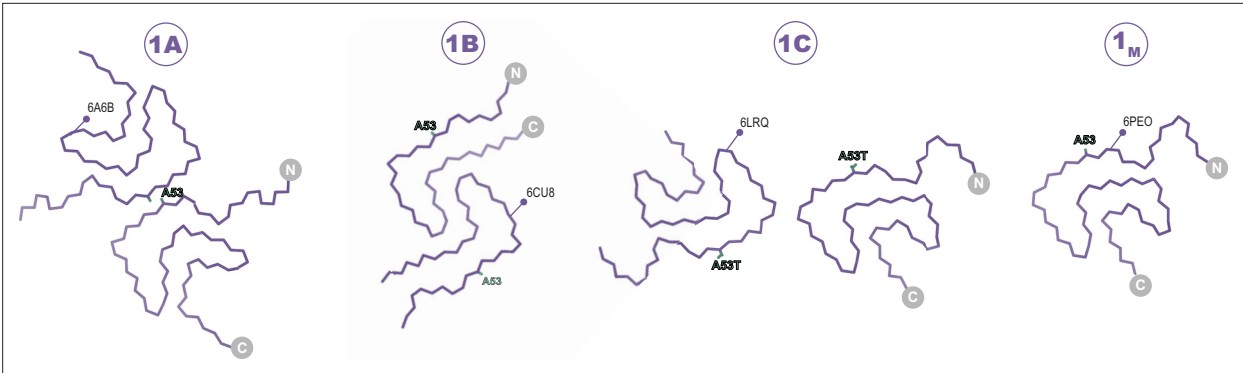

**Figure 2.** The Type 1 polymorphs. Backbone Cα traces showing the four different interface polymorphs of Type 1, including 1B whose fold is not classically Type 1 but whose name is kept for consistency with previous publications. The 1C polymorph is only formed by mutants at the 1A interface (here A53T) and so residue 53 is highlighted in each structure.

protofilament interfaces given different letters (with the exception of polymorph 1B whose name is kept for consistency with previous publications despite itself being a distinct fold, see *Figure 2*). We find that the pH of the aggregation condition is a major but not the only factor in determining which polymorph is formed in vitro. While attempting to reproduce the published conditions for the Type 2 polymorph, we found a new polymorph (here termed Type 5) and a monofilament Type 1 polymorph, providing further evidence that the energetic landscape of α-Syn aggregation is rough with many interconnected local minima. In addition, our findings with cross-pH seeding experiments support the notion that seeding with α-Syn is non-polymorph-specific under conditions of secondary nucleation-dominated aggregation (*Peduzzo et al., 2020*), consistent with difficulties in recreating the in vivo observed fibrils through seeding (*Lövestam et al., 2021*).

## Results and discussion
### The range of accessible polymorphs is controlled by pH

There are more than 60 in vitro amyloid structures of α-Syn deposited in the PDB, representing over a dozen different structural polymorphs. The authors of these structures often try to frame newly observed polymorph(s) in the context of the diseases associated with α-Syn aggregation. However, given so many published structures, an examination of the conditions under which the polymorphs were formed also offers hints into the environmental influences of polymorph selection. In particular, we noted that Type 1 structures were observed almost exclusively above pH 7.0, while Type 2 and 3 structures were observed below pH 7.4 and down to pH 6.5. To systematically analyze the effect of buffer and pH, we studied the structures of α-Syn amyloids of the full-length, N-terminally acetylated protein formed under a variety of conditions. In a first series, we produced α-Syn amyloids in PBS (10 mM PO$_4$, 137 mM NaCl, 2.7 mM KCl) at the physiological pH values of 7.4, 7.0, 6.5, and 5.8.

At pH 5.8 we observed exclusively the Type 3 structures 3B and 3C (*Figures 3 and 4*). Considering the similarities between Type 2 and 3 structures (see below), it is noteworthy that an A-interface polymorph was not observed for Type 3. In its place, we find the 3C polymorph for which the interface contacts are shifted by one residue with respect to 2A and are now defined by a salt bridge between E46 and K58 instead of K45 and E57 (*Figure 5B*). In the previously reported Type 2A and 2B structures solved from a single sample aggregated in PBS at pH 7.4, the major polymorph (83%) was 2A. In our first sample prepared at pH 5.8, we observed 3B as a major polymorph (>90%) with 3C as a minor polymorph. We hypothesized that the absence of a '3A' polymorph and the presence of the 3B as the major polymorph could be related to the ionic strength of the buffer used in aggregation. Therefore, we also pursued cryo-EM structures under similar conditions but with an additional 50 mM or 100 mM NaCl. While we did see a difference in the relative abundance of the 3B and 3C polymorphs in these samples (*Table 1*), '3A' was never detected nor was there an obvious correlation between the NaCl concentration and the 3B:3C polymorph population. In fact, in a second sample prepared under the same original PBS conditions which had yielded an approximately 93%:7% 3B:3C ratio, we

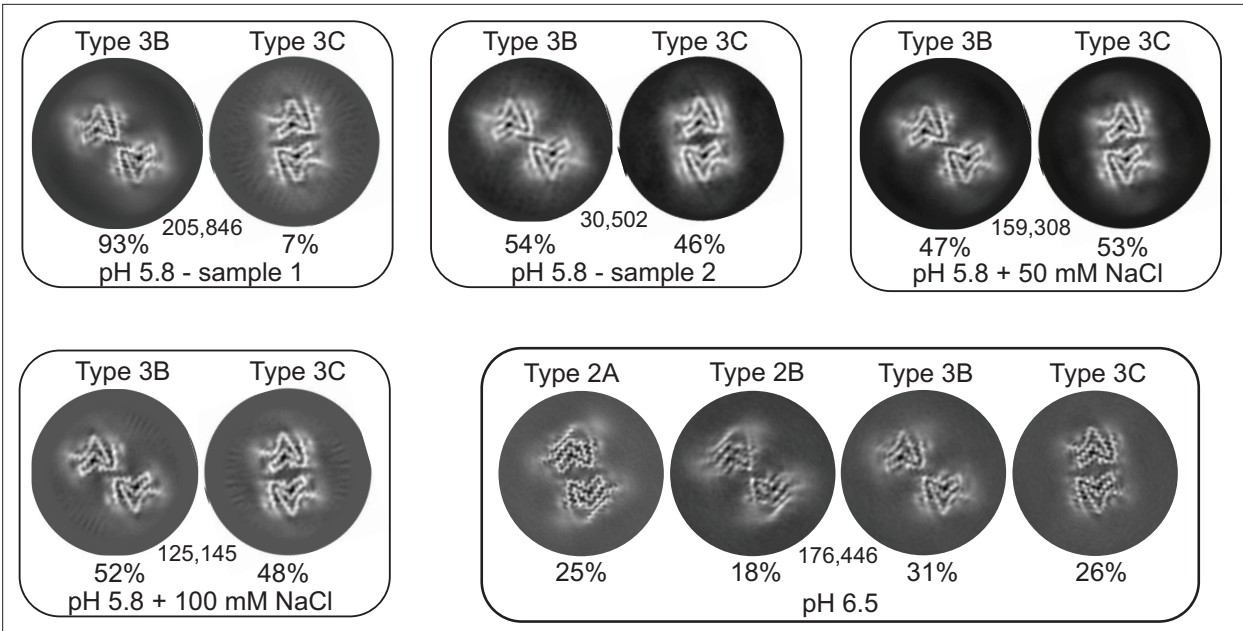

**Figure 3.** 3D classification of particles in fibril samples from various conditions. The 3D classes obtained in RELION for five different fibril samples grown at pH 5.8 and 6.5 in PBS with or without additional NaCl are shown. Each box presents the output classes from one sample with the ratio of the polymorphs determined by the number of particles assigned to each class and the total number of particles used indicated in the center of the box. Classifications were performed with input models for each of the output classes plus one additional cylindrical model (not shown) to provide a 'junk' class for particles that do not fit any of the main classes. The particles that went into the 'junk' class were not included in determining that ratios of particles in the main classes. In the case of the pH 6.5 data, the fibril classes were better separated by filament subset selection in RELION (see Methods and **Table 1**) and the four classes shown represent the output from individual 3D refinements.

encountered yet another ratio of 54%:46%, leading us to the conclusion that stochastic or not well-controlled conditions affect the selection between the 'B' and 'C' interface variants of Type 3 fibrils.

Previous reports of non-acetylated α-Syn fibrils seeded from MSA patient material and grown at pH 6.5 (*Lövestam et al., 2021*) contained a mixture of 2A, 2B, 3B, and 3C polymorphs as well as hybrid 2/3B fibrils. It is therefore noteworthy that in the five independent amyloid samples that we prepared at pH 5.8, none of them contained a detectable level of Type 2 fibrils. However, in the sample prepared at pH 6.5, we did indeed see a mixture of Types 2 and 3 (2A:2B:3B:3C at 25%:18%:31%:26%), suggesting that Type 2 is not accessible below pH 6 (*Figure 3*). The structural basis for the Type 3 preference at lower pH may lie in the protonation state of glutamic acid residues in the vicinity of the C-terminal β-strand (i.e. E83) or of H50, two regions where polymorphs 2 and 3 are most distinct from each other (discussed below). The fact that we observe the same set of polymorphs in the absence of MSA seed material suggests that not only was the seeding described above non-polymorph specific but also that pH was the main polymorph determinant. Seeding experiments have also been performed with CSF of PD patients at pH 6.5, this time with N-terminally acetylated α-Syn, yielding once again the 3B and 3C polymorphs (*Fan et al., 2023*).

In both the MSA and Parkinson's-CSF seeding experiments, the authors also reported on some samples that produced Type 1-like polymorphs which, unlike most in vitro polymorphs, existed primarily as single monofilament fibrils. These are noteworthy because the large majority of in vitro and in vivo α-Syn polymorphs are composed of more than one protofibril, most often with a C2 or pseudo $2_1$ helical symmetry. Therefore, we were surprised to find that in several pH 7.0 aggregation samples (*Table 1*, *Figure 4*), we also observed a Type 1 monofilament fibril polymorph (termed $1_M$), similar to those from the MSA and Parkinson's-CSF seeding experiments and very similar to the Juvenile-onset synucleinopathy (JOS) polymorph (*Lövestam et al., 2021*; *Strohäker et al., 2019*; *Burger et al., 2021*; *Figure 6*, discussed below). This again indicates that seeding experiments may be influenced by environment-dictated polymorph formation. Also puzzling to us is that we could not produce Type 2 polymorphs at pH 7.0 or 7.4 despite previous reports of 2A and 2B being formed under these conditions (*Guerrero-Ferreira et al., 2019*; *Frieg et al., 2021*). In yet another of the pH 7.0 aggregation

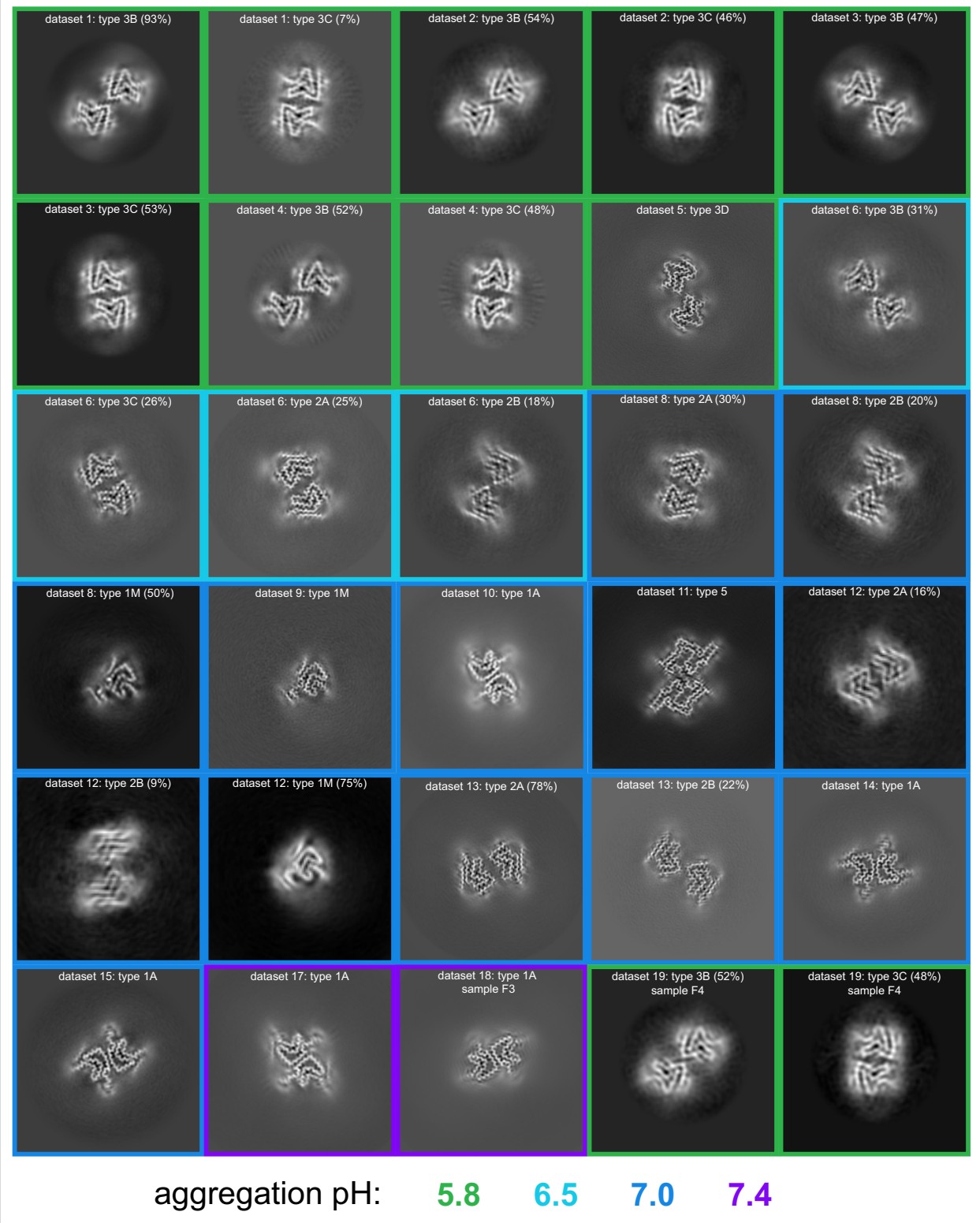

**Figure 4.** pH-dependence of observed α-synuclein (α-Syn) polymorphs. An overview of all of the cryo-electron microscopy (cryo-EM)-resolved polymorphs in this study. The images represent a 9.5 Å slice (about two layers of the amyloid) of the 3D map projected in the Z direction. The 3D maps were obtained either from 3D classification or 3D refinement in RELION. The datasets are ordered and numbered as in **Table 1**. The colored borders indicate the pH at which the fibrils were grown. For datasets from which multiple polymorphs were resolved, the relative percentage of particles corresponding to each polymorph is indicated in parentheses.

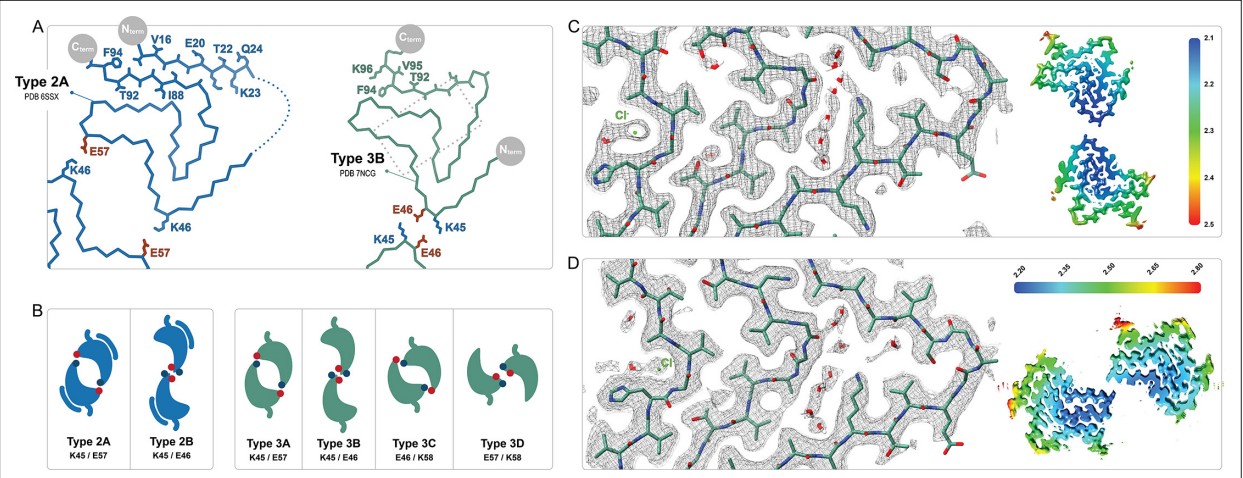

**Figure 5.** Comparison of the Type 2 and 3 polymorphs. (**A**) Protofilaments of polymorphs 2A (blue) and 3B (green) are depicted as Cα traces. The side chains are included on the C-terminal β-strand and the N-terminal segment in the 2A structure in order to highlight the inside-out flipped orientations of their C-terminal β-strands. The dashed box indicates the region highlighted in panels **C** and **D**. (**B**) The two interfaces of Type 2 and the 4 interfaces of Type 3 polymorphs are shown as a cartoon schematic. The charged interfacial residues are indicated by blue/red dots for K/E and listed below each interface. The C-term is indicated as a short tail and Type 2 has the extra N-terminal segment. (**C**) A close-up view of the Type 3B structure (PDB:8PIC) and (**D**) Type 3D (E46K) structure (PDB:8PJO) and their EM density, showing the shared set of immobilized water molecules and strong density that has been modeled as a chloride ion. The local resolution maps for each structure are shown to the right with the color scale indicating the resolution range in Å.

samples we found a completely new polymorph which we have termed Type 5 (*Figure 7*). Apart from its novel fold, it is unusual in the number of residues that are visible/ordered in the structure, starting from the acetylated N-terminal methionine (buried at the dimer interface) up until residue 96. Interestingly in a recent BioRxiv manuscript (*Burger et al., 2021*) (for which the coordinates are currently on hold), a similar polymorph is found upon seeding with MSA and PD material, again suggesting that the environment could play a role in the selection of polymorphs (*Li et al., 2018*).

In the original reports of Type 2 structures (2A and 2B), fibrils were formed by non-acetylated α-Syn at pH 7.5 in Tris buffer as well by three other constructs at pH 7.0–7.3 in DPBS: (i) the E46K mutant, (ii) phosphorylation at S129, and (iii) an N-terminally acetylated α-Syn. Therefore, we were somewhat surprised that after measuring cryo-EM data for six independent fibrillization samples at pH 7.0 and 7.4 in PBS or Tris (*Table 1*), we could not detect any Type 2 polymorphs. Instead, we found that at pH 7.0 in PBS, the samples exhibited a multitude of polymorphs including one having mostly flat (not twisted) fibrils of variable width and the other two each yielding a new polymorph (5A and $1_M$). However, in Tris buffer at pH 7.0 we did produce polymorph 1A as the major species. Considering the strong temperature dependence of the Tris p$K_a$, it is worth noting that our Tris buffer was prepared for pH 7.0 at the fibrillization temperature of 37°C. Returning to PBS but at pH 7.4, polymorph 1A was the dominant species in both samples that we prepared, even though one was seeded by 5% Type 3 fibrils that had grown at pH 5.8 (the latter discussed in more detail below). While in our hands 1A is the preferred polymorph above pH 7.0, regardless of the buffer used, we also observed a higher degree of sample variability in the samples grown at pH 7.0 and 7.4 (*Table 1*, *Figure 4*). In order to further explore the variability/reproducibility of the pH 7.0 polymorphs, we measured cryo-EM data for six more independent fibril samples for a total of 10 samples, all aggregated independently at pH 7.0 and originating from four different batches of purified α-Syn (datasets 7–16, *Table 1*). In three of these pH 7.0 samples we did finally reproduce the type 2A/B polymorphs (datasets 11–13), two of which also had a significant amount of Type $1_M$ fibrils. In yet two others datasets (14–15) we observed only Type 1A. Several samples contained predominantly non-twisted fibrils and could therefore not be used for helical reconstruction. In most samples, we also observed fibrils that were thinner or wider than the known polymorphs, but these were not pursued due to a poor homogeneity between fibrils, lack of twist, or an insufficient number of specimens and so were excluded either during manual picking or later in 2D and 3D classification.

**Table 1.** Cryo-electron microscopy (cryo-EM) samples analyzed for this manuscript.

| Dataset | Construct | Batch | Aggregation condition | pH [‡] | Polymorph(s) | Relative abundance[§] | Refinement stage |
|---|---|---|---|---|---|---|---|
| 1 | Ac-1–140* | 2 | PBS[†] | 5.8 | 3B[¶]:3C | 93%:7% | 3B, see *Tables 2* : 3C, 3D classification |
| 2 | Ac-1–140 | 6 | PBS | 5.8 | 3B:3C | 54%:46% | 3D classification |
| 3 | Ac-1–140 | 3 | PBS+50 mM NaCl | 5.8 | 3B:3C | 47%:53% | 3C, see *Table 2* : 3B, 3D classification |
| 4 | Ac-1–140 | 4 | PBS+100 mM NaCl | 5.8 | 3B:3C | 52%:48% | 3D classification |
| | Ac-1–140 (E46K) | | PBS+50 mM NaCl | 5.8 | 3D | | See *Table 2* |
| 6 | Ac-1–140 | 7 | PBS | 6.5 | 2A:2B:3B:3C | 25%:18%:31%:26% | Filament subset selection followed by 3D refinement to 3.51 Å, 4.73 Å, 4.56 Å, 4.06 Å respectively |
| 7 | Ac-1–140 | 4 | 20 mM Tris, 140 mM NaCl | 7.0 | 1A | | 3D refinement 3.65 Å |
| 8 | Ac-1–140 | 4 | PBS | 7.0 | 5A | | See *Table 2* |
| 9 | Ac-1–140 | 5 | PBS | 7.0 | 1$_M$ | | See *Table 2* |
| 10 | Ac-1–140 | 5 | PBS | 7.0 | Non-twisted** | | |
| 11 | Ac-1–140 | 7 | PBS | 7.0 | 2A:2B:1$_M$ | 30%:20%:50%[‡‡] | Filament subset selection followed by 3D refinement to 4.2 Å, 4.9 Å and 4.8 Å respectively |
| 12 | Ac-1–140 | 7 | PBS | 7.0 | 2A:2B:1$_M$ | 16%:9%:75%[‡‡] | Filament subset selection followed by 3D refinement to 5.4 Å, 6.4 Å and 5.8 Å respectively[††] |
| 13 | Ac-1–140 | 8 | PBS | 7.0 | 2A:2B | 78%:22% | Filament subset selection - see *Table 2*[††] |
| 14 | Ac-1–140 | 8 | PBS | 7.0 | 1A | | 3D refinement to 2.95 Å |
| 15 | Ac-1–140 | 8 | PBS | 7.0 | 1A | | 3D refinement to 3.94 Å |
| 16 | Ac-1–140 | 8 | PBS | 7.0 | Non-twisted and clumped** | | |
| 17 | Ac-1–140 | 4 | PBS | 7.4 | 1A | | 3D refinement to 3.73 Å |
| 18 (F3) | Ac-1–140 | 5 | PBS +5% pH 5.8 seeds | 7.4 | 1A | | 3D refinement to 3.50 Å |
| 19 (F4) | Ac-1–140 | 7 | PBS +5% pH 7.4 seeds | 5.8 | 3B:3C | 52%:48% | 3D classification |

*All constructs are full-length, N-terminally acetylated (Ac).

[†]PBS is a 10 mM phosphate buffer solution with 137 mM NaCl and 2.7 mM KCl.

[‡]The pH was adjusted after dissolving the PBS tablet (Sigma-Aldrich) by the addition of HCl.

[§]In samples for which more than one polymorph could be identified by 3D classification or filament subset selection in RELION.

[¶]Underlining indicates data that was used for 3D refinement of the deposited maps and for building the deposited coordinates.

**This data could not be analyzed by helical reconstruction as only non-twisted fibrils were present.

[††]Filament subset selection was run after 2D classification of the entire set of auto-picked particles. The identified filament classes were individually 2D classified and these classes used to create an initial model with relion_helix_inimodel2d which was used for automated 3D refinement. In the case of dataset 13 followed by CTF refinement and Bayesian polishing in RELION.

[‡‡]Due to the small size of these datasets, the relative abundances are not likely to be precise.

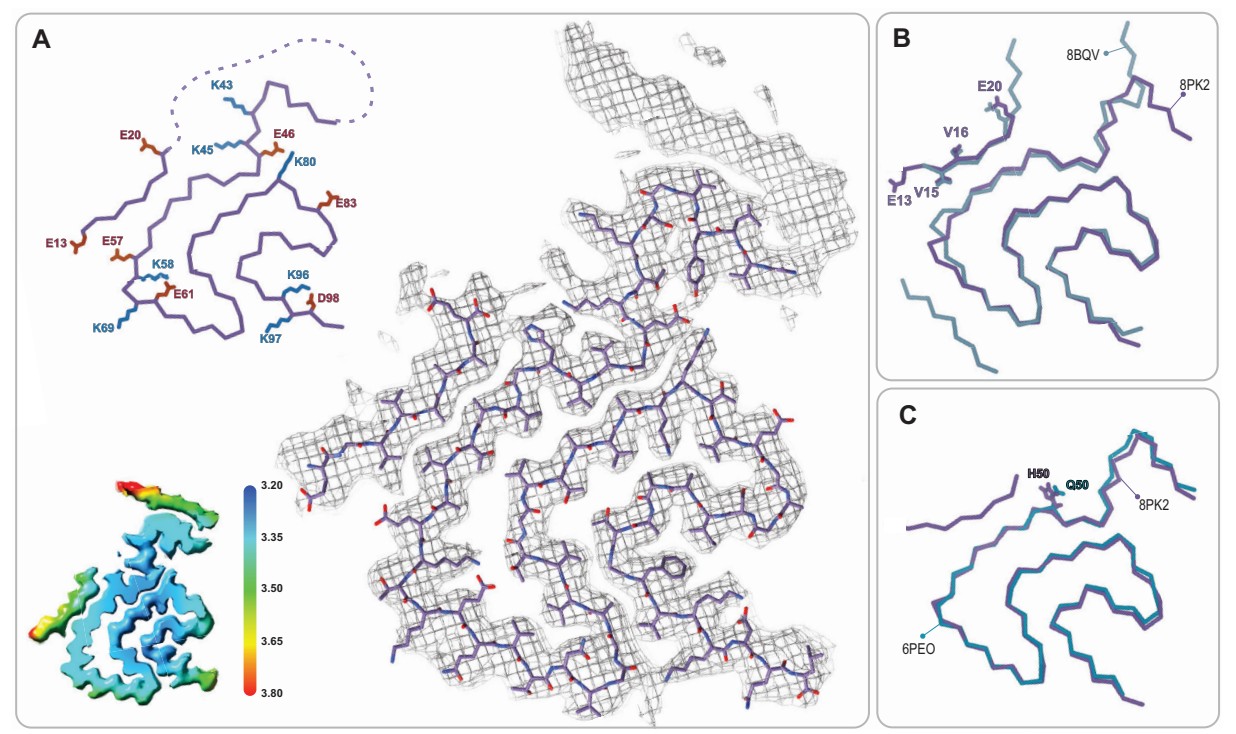

**Figure 6.** The juvenile-onset synucleinopathy (JOS)-like Type 1$_M$ polymorph. (**A**) The Type 1$_M$ monofilament structure (PDB:8PK2) overlaid with its EM density (including the unmodeled density). The Cα trace is also shown on the left with all K and E side chains indicated as well as the local resolution map for the density with color scale showing the resolution range in Å. (**B**) An overlay of the Cα trace of the Type 1$_M$ and the JOS polymorph (PDB:8BQV) from which the identity of residues 13–20 in the N-terminal strand were assigned. (**C**) An overlay of the Cα trace of the JOS-like Type 1$_M$ to the 1$_M$ structure of the H50Q mutant (PDB:6PEO). The structural alignments were done by an LSQ-superposition of residues 51–66.

## On the structural heterogeneity of Type 1 polymorphs

27 structures of Type 1 polymorphs have been reported, mostly in the pH range of 7.0–7.4 in distinct buffers including Tris or phosphate and comprised of full-length α-Syn, N-acetylated α-Syn, shorter constructs (i.e. 1–103, 1–122), familial mutants (i.e. A53T, A53E, H50Q, and a seven residue insertion associated with JOS), patient-derived material (i.e. MSA and JOS), and fibrils that were seeded by MSA, PD, and dementia with Lewy bodies material (including brain or CSF). The Type 1 polymorphs display a large structural heterogeneity both at the tertiary structure level (*Figure 8*) and at the quaternary level, having distinct protofilament interfaces or even existing as a monofilament fibril (*Li et al., 2018*; *Li, 2018*; *Guerrero-Ferreira et al., 2018*; *Sun et al., 2020*; *Ni et al., 2019*; *Schweighauser et al., 2020*; *Lövestam et al., 2021*; *Sun et al., 2023*; *Zhao et al., 2023*; *Zhang et al., 2023*; *Frieg et al., 2022*; *Yang et al., 2023*; *Fan et al., 2023*). This can be rationalized in part, for example, with the familial mutants A53T, A53E, and H50Q which all have been found in polymorph 1C with its smaller protofilament interface (residues 59–60) compared to 1A (residues 50–57). The mutants lie at the 1A interface, destabilizing it and favoring the smaller interface of 1C or even the monofilament 1$_M$ (illustrated in *Figure 2*). In the context of how buffer conditions influence the polymorph structure, it is interesting to mention that in the presence of phosphate, the K58 side chain is often facing out toward the 1A protofilament interface, thereby forming a triad with K43 and K45 of the other protofilament to surround some extra density that is interpreted to be phosphate (*Figure 8*). This constellation of lysine residues is found in about two-thirds of the dozen structures of amyloids grown in the presence of phosphate. In contrast, in phosphate-free conditions (such as Tris, PIPES, or unbuffered) K58 is facing into the core in about two-thirds of a dozen structures, forming a salt bridge/hydrogen bond with E61. Since in the presence and absence of phosphate both K58 orientations have been documented, other buffer details such as pH may also be relevant for this and other variations. Furthermore, it points to a rather shallow amyloid energy hypersurface that can be altered by subtle changes in the environment.

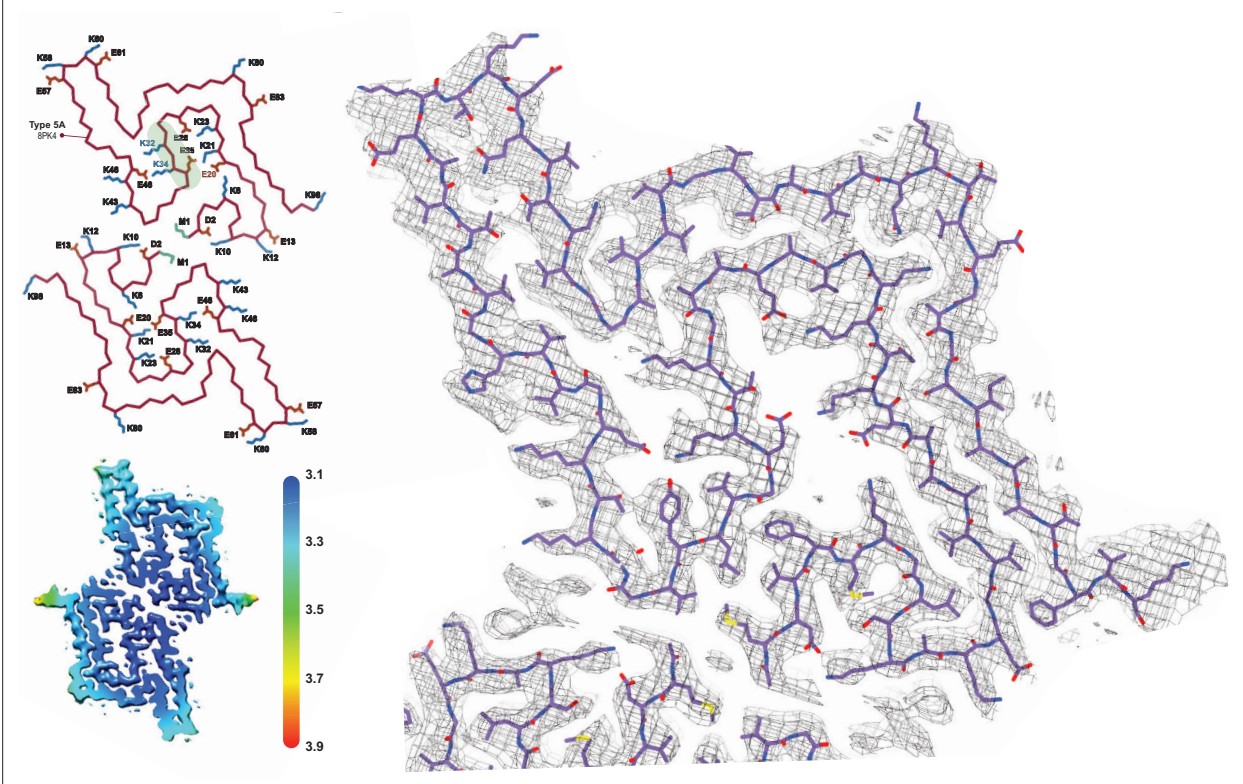

**Figure 7.** The Type 5 polymorph. The structure of a Type 5 protofilament (PDB:8PK4) overlaid with its EM density. The Cα trace for the two filaments of the Type 5A fibril is also shown to the left. All of the charged (E/D/K) residues and the acetylated N-terminal Met are labeled and the polar segment that dissects the two cavities is shaded green in the upper chain. The local resolution map is also shown with the color scale indicating the resolution in Å.

In the context of polymorph 1 and the interfacial lysine triad discussed above, it is worth mentioning the structure of the disease-relevant JOS brain-derived polymorph. In this polymorph, the 1A protofilament interface is disrupted by an additional β-strand comprising the N-terminal residues E12-E25, yielding a monofilament structure that we term Type $1_M$ as well as a double protofilament structure with a unique protofilament interface. Consequently, the lysine triad of 1A is lost (with K58 still facing outward), but now K43 and K45 together with K21 and K23 of the N-terminal additional β-strand embrace a small molecule that is likely negatively charged, yielding overall a similar local structural arrangement around K43 and K45 (and H50). An attempt to produce the JOS polymorphs in vitro using a mixture of wild-type α-Syn and variant containing the JOS-associated seven-residue insertion MAAAEKT after residue 22 yielded instead polymorph 1A structures with the typical protofilament interface (*Yang et al., 2023*). However, in our structure-buffer relationship screening, a JOS-like $1_M$ polymorph containing the additional N-terminal strand of the JOS polymorph was obtained with wild-type N-terminally acetylated α-Syn. There is only a small difference in the conditions, pH 7.4 versus pH 7.0 in PBS, that lead to the 1A versus the JOS-like $1_M$ polymorphs, again illustrating the shallow energy surface between the polymorphs and the critical role of the buffer conditions in polymorph selection. The in vitro prepared JOS-like $1_M$ polymorph superimposes with the JOS purified material almost entirely except for the aforementioned K58 side chain facing inward in the former and outward in the latter, and the unassigned peptide density in the latter (*Figure 6B*). The cryo-EM map for the JOS-like $1_M$ fibrils also has an additional sausage-like density, likely for a β-strand, but it was not used for model building because it lacked enough features (*Figure 6A*).

## On the structural heterogeneity of Type 2 and 3 structures

The structures of Type 2 and 3 polymorphs share some similarities, however, there are significant differences that justify their assignment to unique polymorph classes (*Figure 5AB*). First, there is a displacement of the structural fragment comprising residues V52-T72 which appears to move on the

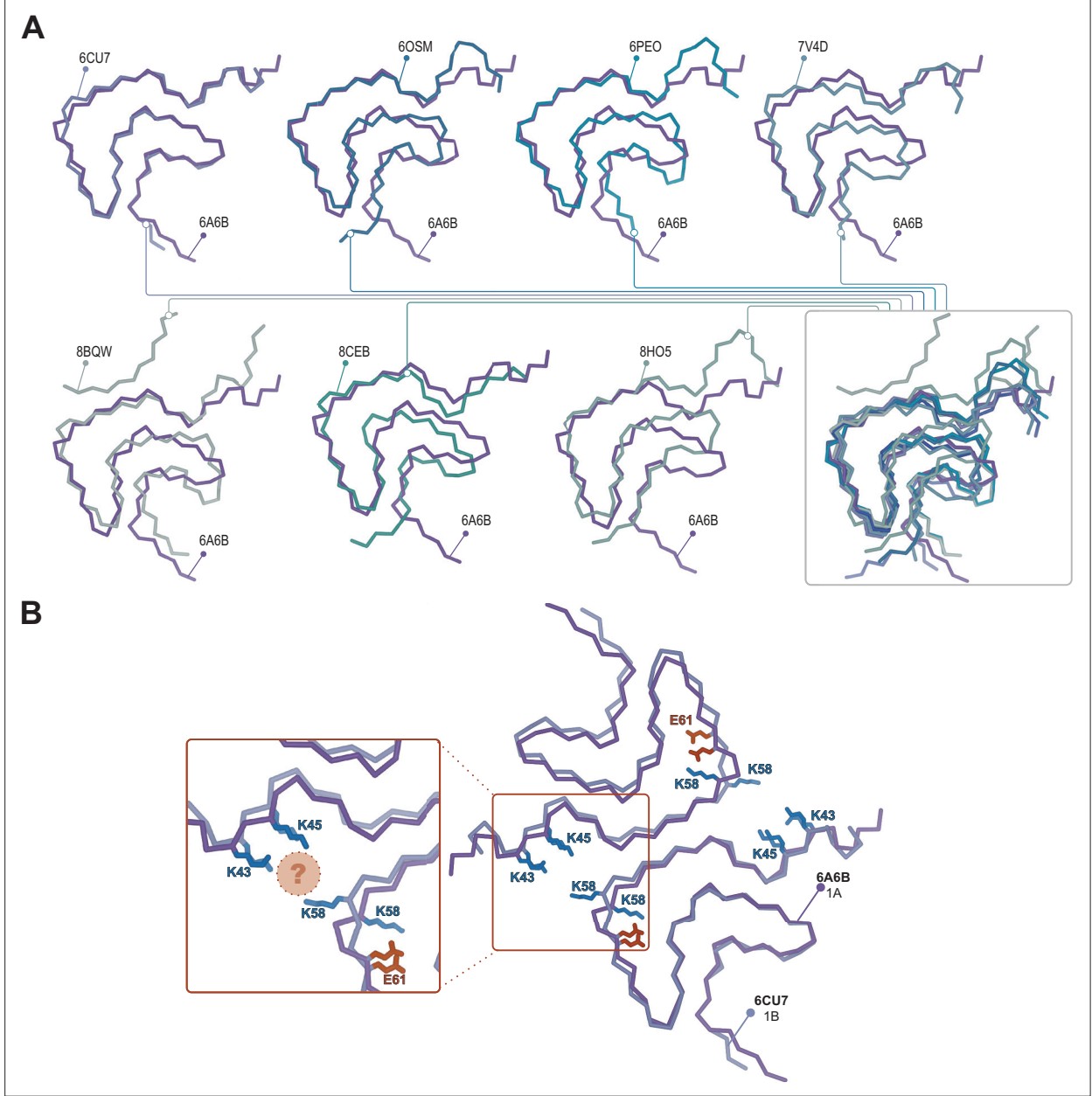

**Figure 8.** Structural variability within the Type 1 polymorphs. (**A**) Seven pairwise overlays of the wild-type 1A structure (PDB:6A6B) with other wild-type and mutant structures depicting the range of structural variability that is found in the Type 1 fold. The structural alignments were done by an LSQ-superposition of the Cα atoms of residues 51–66. The other Type 1 structures from left to right, top to bottom are wild-type 1A without buffer (PDB:6CU7), N-terminally acetylated wild-type 1A residues 1–103 (PDB:6OSM), the H50Q mutant $1_M$ without buffer (PDB:6PEO), wild-type 1A in Tris-buffered saline (PDB:7V4D), the patient-derived juvenile-onset synucleinopathy (JOS) $1_M$ polymorph (PDB:8BQV), mix of wild-type and seven residue JOS-associated insertion mutant 1A in PBS (PDB:8CEB) and N-terminally acetylated wild-type $1_M$ seeded from Parkinson's disease (PD) patient cerebrospinal fluid (CSF) in PIPES with 500 mM NaCl. (**B**) Same overlay shown at the top right in **A**, showing the triad of lysine residues often formed at the interface in the presence of phosphate (PDB:6CU7) compared to the inward-facing orientation of K58 in the absence of phosphate (PDB:6A6B). The location of often observed density, thought to be $PO_4^{2-}$, is indicated with the pink sphere.

G51 and G73 hinges. The turn with residues G73-A78 is again similar between the two structures but beyond this, there is an 'inside-out' rearrangement of the C-terminal β-strand (A85-G86-S87-I88-A89-A90-A91-T92) such that only in the Type 2 polymorph can the N-terminal fragment (V15-K21) form an additional β-strand packed against the β-strand A85-A91 (*Guerrero-Ferreira et al., 2018*). However, also within Type 2 and 3 polymorphs exists a large structural heterogeneity: some polymorph 3 structures have an 'inside-out' C-terminal β-strand but still without the N-terminal fragment present. In

terms of the interface variants, there is a difference between our Type 3B structure and that of from the MSA-seeded Type 3B (*Lövestam et al., 2021*). Our data shows the characteristics of a pseudo-$2_1$ helical symmetry (ca. 179.5° twist and 2.35 Å rise) when refined under C1 symmetry with a ca. –1° twist and 4.7 Å rise (*Figure 9*) while the MSA-seeded Type 3B structure (PDB entry 7NCG) has a C2 symmetry (–0.95° twist 4.75 Å rise).

The interfaces present in the 2A and 3C polymorphs also appear to be similar at first glance, consisting of a pair of symmetric salt bridges arranged to create a large solvent-filled cavity at the interface. However, a close inspection reveals that they are different, comprising neighboring sets of E/K residues: K45/E57 in 2A and E46/K58 in 3C. This is remarkable because there is no obvious reason that the 'C' interface is preferred over the 'A' interface. An example of a 3A polymorph has been reported but it is in the context of the E46K mutant (*Boyer et al., 2020*) and formed in an unbuffered solution. The familial PD mutant α-Syn (E46K) (*Zarranz et al., 2004*) has been extensively studied and the structures of fibrils of this mutant have been reported at pH 7.5 in Tris (*Zhao et al., 2020a*), at pH 7.4 in PBS (*Guerrero-Ferreira et al., 2019*) and as mentioned above, unbuffered in 15 mM tetrabutyl-phosphonium bromide (*Boyer et al., 2020*). The Tris condition produced a Type 4 polymorph, the PBS a Type 2A, and the unbuffered condition a Type 3A. Interestingly, since the 2A and 3A polymorphs share the same dimer interface with symmetric K45/E57 salt bridges it is clear that the 'A' interface is not mutually exclusive with the Type 3 polymorph. However, it is still not clear as to why Type 3B and 3C polymorphs, both with E46 in their salt-bridged dimer interfaces, are preferred for the wild-type protein but we suspected that it was only through the elimination of E46 that the E46K mutant could access the 3A polymorph. To test this further, we produced E46K fibrils at pH 5.8, the condition where until now only Type 3 has been observed and uncovered yet another Type 3 polymorph with a new interface termed 3D formed by an E57/K58 salt bridge (*Figure 5B*). This was surprising considering the previously mentioned Type 3A structure of this mutant and suggests that the Type 3 fibrils formed at pH 5.8 are somehow distinct from those formed, until now only by mutants, above pH 7. While researching deposited α-Syn structures we found another Type 3D polymorph (PDB entry 7WO0, not yet published) that is formed by the A53T mutant in the presence of 20 mM $CaCl_2$ at pH 7.6. Finally, the coexistence of polymorphs 2A/2B and 3B/3C at pH 6.5 indicates either that they all have a similar thermodynamic stability or that some of them are kinetically trapped, the latter possibility having been observed in one case of two distinct α-Syn polymorphs grown in a single condition (10 mM Tris, 0.02% $NaN_3$, pH 7.0 at 37°C) (*Pálmadóttir et al., 2023*).

## The Type 5 structure

Apart from its novel fold, polymorph 5A is unusual in the number of residues that are visible/ordered in the structure, starting from the acetylated N-terminal methionine (buried at the dimer interface) up until residue E96. As is often the case, there are shared structural motifs between Type 5 and other α-Syn polymorphs. In particular, residues 47–72, comprising a two β-strand motif with a turn in between, overlay very well with the polymorph formed by a Y39 phosphorylated α-Syn (*Zhao et al., 2020b*) and to a lesser extent in Types 2, 3, and 4 as well. However, the first 35 residues are distinct in that they are usually disordered (Types 1 and 3), or only a segment thereof binds to the core in Type 2 or polymorph $1_M$. Within this segment, there are seven lysine and five glutamic/aspartic acid residues and five of the lysine residues are buried inside the structure and form salt bridges with buried glutamic/aspartic acid residues. This yields two large cavities separated only by the very polar segment G31-K32-T33-K34-E35-G36, and it is likely that these two cavities are filled with water. In comparison to the other known polymorph structures the size of the cavities and the presence of five internal salt bridges are unique. Only polymorph 1 shows a small significant cavity which as discussed above usually appears in the presence of phosphate buffer. However, even the more compact Type 3 structures which do not have any obvious cavities contain networked channels of waters that are shared between the wild-type Type 3B and E46K Type 3D structures (*Figure 5C and D*). The occurrence of the Type 5A polymorph thus reflects again on the promiscuity of α-Syn polymorph formation, the influence of slight changes in buffer conditions, and more specifically the potential role of lysine residues in polymorph selection. It is also worth mentioning that in the 10 independent aggregations that we analyzed at pH 7.0, the Type 5 structure was found in only one sample, and then, in only 10–20% of the observed fibrils in the sample. The remainder of the fibrils in the sample were non-twisted and therefore could not be analyzed by 3D reconstruction.

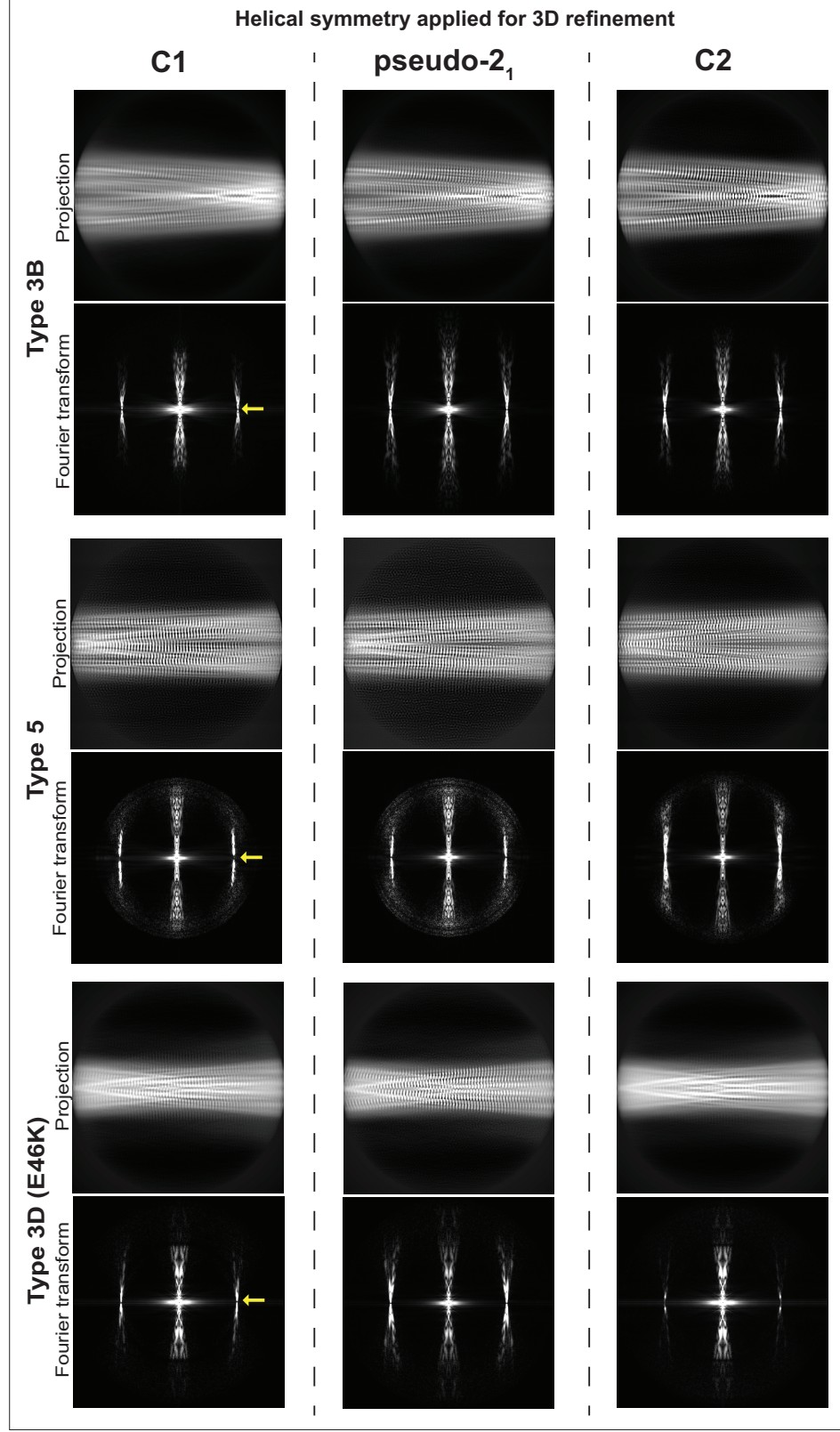

**Figure 9.** Determination of helical symmetry in Type 3B, 3D, and 5A polymorphs. 2D projections and their Fourier transforms for three different structures (rows) refined with three different types of symmetry (columns). Each of the structures studied was refined as far as possible in a C1 symmetry with a ca. –1° twist and 4.7 Å rise in order to examine the higher order symmetry expected to be present: either C2 with a 4.7 Å rise or a pseudo-2₁ helical

*Figure 9 continued on next page*

*Figure 9 continued*

symmetry with ca. −179.5 twist and 2.35 Å rise. A comparison of the projections of the refined volumes indicated that all three of these structures have a pseudo-$2_1$ helical symmetry. This can be seen in the C1 projections which lack a mirror symmetry down their middle and in their Fourier transforms which lack an n=0 Bessel function meridional peak in the layer line at 1/4.7 Å (location marked with a yellow arrow).

## The buffer environment can dictate polymorph during seeded nucleation

While for many amyloids, seeded growth results in replication of the structural polymorph of the seeds, in at least one instance, the seeding with MSA brain-derived fibrils failed to reproduce the seed polymorphs (*Lövestam et al., 2021*). One explanation could be the absence of the as yet unknown cofactor that appears to be bound in MSA polymorphs. However, as we have observed, the buffer environment/condition may play such an important role in polymorph selection that we wondered whether, for pure in vitro seeding experiments with in vitro-derived seeds, it would be the environment or the seed which dictates the resulting polymorph. To address this, we used fibrils of polymorph 3B/3C produced at pH 5.8 as seeds (5%) in seeding experiments at pH 7.4 and as a control at pH 5.8. The inverse experiment was also carried out with fibrils of the Type 1 polymorph produced at pH 7.4 used as seeds (5%) in seeding experiments at pH 5.8. In both cases, control seeding experiments were performed at the pH of the original seed fibrils. For these analyses, we monitored the effect of seeding on the kinetics of fibril formation using thioflavin T (ThT) fluorescence (*Rogers, 1965*; *LeVine, 1999*) and probed the structures of the resulting seeded fibrils by cryo-EM.

In terms of kinetics, the presence of seed fibrils consistently accelerated the aggregation of α-Syn (*Figure 10A*). However, in the cryo-EM analyses of the fibrils produced at pH 5.8 with pH 7.4 seeds, only the polymorphs 3B and 3C are detected (*Figure 10B*) and in the inverse case, with fibrils produced at pH 7.4 with pH 5.8 seeds only polymorph 1A is detected (*Figure 10B*). We were unable to detect any traces of the seed polymorphs in either of the cross-seeded aggregations despite the fact that 5% seeds were used. Limited proteolysis-coupled mass spectrometry (LiP-MS) (*Schopper et al., 2017*; *de Souza and Picotti, 2020*), a quantitative technique that enables the detection of nearly single residue resolution structural differences/alterations in proteins in bulk solution, confirmed these findings (*Figure 10C*). In our LiP-MS analysis, we digested the fibril samples with the sequence-unspecific protease proteinase K for a limited amount of time to generate structure-specific proteolytic fingerprints and then identified and differentially quantified peptides between the different fibril states by MS analysis. To obtain single amino acid resolution in our comparison of the structural features of different types of fibrils, we first assigned a score accounting for the extent of change between samples and a measure of statistical significance to each detected peptide and then aligned the peptides with overlapping regions to calculate the mean score value for each amino acid (see Methods for details). Thus, larger differences between two structures yield higher per-residue scores. LiP-MS analysis revealed that fibrils generated at pH 7.4 (F3) and pH 5.8 (F4), in the presence of unfragmented fibrillar seeds generated at pH 5.8 and pH 7.4 respectively, are on average more similar in their structure to the fibrils generated without seeds at pH 7.4 (F1) and pH 5.8 (F2), respectively (*Figure 10C*). These data suggest that fibrillar conformations resulting from seeding monomeric α-Syn with unfragmented fibrillar seeds are driven by pH and not by the structure of the seed.

Amyloid aggregation can be accelerated by two seed-based processes: (i) fibril elongation and (ii) secondary nucleation. In fibril elongation, the incoming protein adopts the conformation of the seed by attaching itself to the end of the fibril. In this process of growth acceleration, which is proportional to the number of fibril ends, the polymorph structure is expected to be preserved (*Jarrett and Lansbury, 1993*; *Jarrett and Lansbury, 1992*). In the secondary nucleation process, de novo aggregation occurs on the sides of the fibril and therefore the kinetics of secondary nucleation is proportional to the mass of the fibril (*Knowles et al., 2007*). In the case of α-Syn, the secondary nucleation mechanism has been proposed to be based on the interaction of the positively charged N-terminal region of monomeric α-Syn and the disordered, negatively charged C-terminal region of the α-Syn amyloid fibrils (*Kumari et al., 2021*). The interaction is transient but causes a further unfolding of the monomeric α-Syn while increasing its local concentration and possibly aligning it in a way that promotes aggregation. Since all known amyloid polymorphs of α-Syn are disordered in their C-terminal regions, we propose that secondary nucleation in α-Syn aggregation is generally polymorph unspecific. This

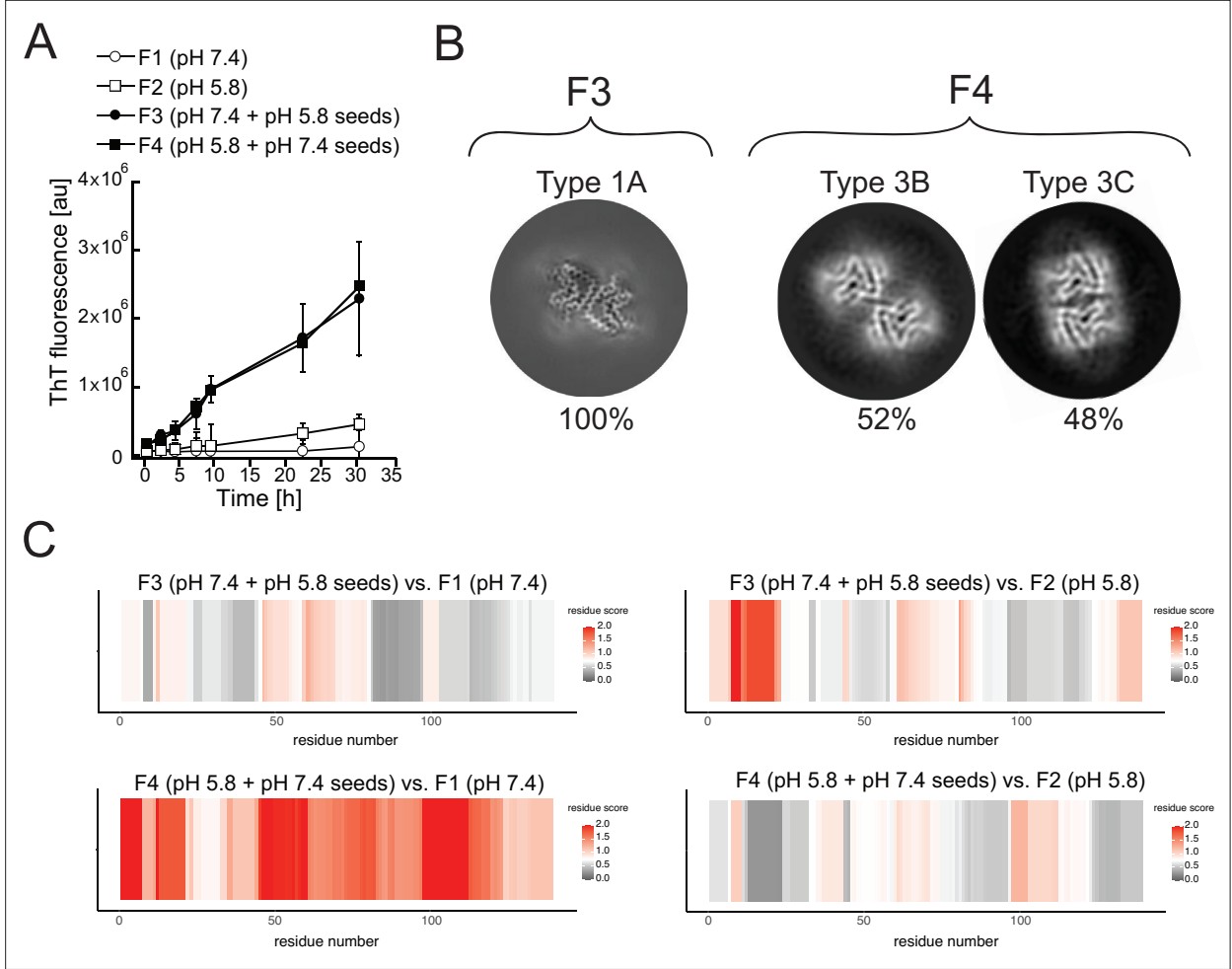

**Figure 10.** Cross-seeding does not preserve the seed polymorph. Fibrils produced at pH 7.4 (F1) and pH 5.8 (F2) were used as seeds to generate new fibril samples with fresh α-synuclein monomer: at pH 7.4 with 5% unfragmented pH 5.8 seeds (**F3**) or at pH 5.8 with pH 7.4 seeds (F4). (**A**) The effect of cross-seeding with fibrils produced in a different pH from that of the aggregation condition is shown in the aggregations kinetics, as monitored by thioflavin T (ThT) fluorescence. The average values of three independent aggregation measurements are plotted with error bars representing their standard deviations. (**B**) Cryo-electron microscopy (cryo-EM) analyses of the seeded samples yielded a single 3D class representing polymorph 1A (here depicted as a Z-section) for the F3 fibrils and two classes representing Types 3B and 3C for the F4 fibrils. (**C**) Limited proteolysis-coupled mass spectrometry (LiP-MS) analysis showing the differences in fibrillar structures in bulk solution. The comparison of seeded fibrils (F3 and F4) with F1 is presented on the left, while the comparison of seeded fibrils with F2 is presented on the right. Differences in structures per residue are plotted along the sequence of α-synuclein in a form of scores ($-\log_{10}$(p-value) × |$\log_2$(fold change)|). The more intense the red color, the greater the difference between the two structures in each region. Gray indicates no significant differences, while white indicates the significance threshold corresponding to ($-\log_{10}$(0.05) × |$\log_2$(1.5)|).

conclusion is consistent with those of *Peduzzo et al., 2020*, who showed that at lower seed concentrations under which secondary nucleation dominates over fibril elongation, the environment dictates the resulting polymorph. It is worth noting that unfragmented fibrils will likely favor secondary nucleation, while with shorter fibrils and more available ends per unit mass, elongation could dominate.

In addition to the α-Syn seeding experiments discussed until now, *Strohäker et al., 2019*, used NMR to analyze the results of PMCA using both PD and MSA seed material in PBS buffer at pH 7.4 in the presence of 1% Triton X-100. Both samples yielded indistinguishable polymorphs. Hence, it appears that their protocol amplified the seeds via secondary nucleation for which a single polymorph can be expected and whose identity is dependent on the environment; here that includes 1% Triton X-100 and 5% wt/vol brain homogenate. In another PMCA study that is published currently only on the BioRxiv server, *Burger et al., 2021*, use both MSA and PD patient-derived material to produce a new polymorph which they name Type 6. A similar Type 6 fold was obtained irrespective of whether

the starting material was from an MSA or PD patient brain, however protofilaments were arranged in a disease-specific manner. Interestingly, the Type 6 fold resembles the Type 5 that we found at pH 7.0, deviating however in its three different protofilament interfaces yielding polymorphs 6A, 6B, and 6C as well as extra density that the authors have assigned to the C-terminal residues. Without the deposited coordinates for the Type 6 structures it is difficult to ascertain how similar these polymorphs are to Type 5, however, their obvious similarity suggests that the environment of the PMCA seeding could have played a role in determining the amplified polymorph. Here, the environment includes the brain homogenate at 2% wt/vol and the PMCA buffer: 150 mM KCl, 50 mM Tris, pH 7.5 (*Burger et al., 2021*) at 37°C, which considering the temperature dependence of the Tris p$K_a$ could actually be close to pH 7 and thus similar to the buffer conditions that produced Type 5. However, it must also be stated that in addition to the cryo-EM structures (*Burger et al., 2021*), previous cell biological and biochemical support for polymorph-specific seeding of the MSA and PD patient-derived material under the same buffer conditions has been reported (*Van der Perren et al., 2020*).

The general impact of the environment for seeding assays was recently investigated in an RT-QuIC assay for which 168 different conditions were tested to find those optimal for α-Syn aggregate detection (*Martinez-Valbuena et al., 2022*). The results indicate that seeding experiments with α-Syn can be very sensitive to the environment and that it is a multifactorial problem where there is not a single optimal pH or optimal salt condition. Another recent report compared the biological activity of (non-seeded) in vitro fibrils versus those amplified from Lewy body diseased brains upon injection into mice brains, with the finding that only the latter faithfully models key aspects of Lewy body disease (*Uemura et al., 2023*). The fact that a pH of 7.4 was used during the seeding process further suggests that there are other environmental factors at play in polymorphs selection. Considering the importance of reproducing the disease-relevant polymorphs and the often irreproducibility of amplification assays it is apparent that approaches need to be found that can establish robust methods for α-Syn polymorph-specific amplification. In addition to finding appropriate buffer conditions, we propose that any approach that reduces or eliminates secondary nucleation is worth pursuing. This could be potentially achieved by using shorter, fragmented fibrils as seeds thereby increasing the fibril end-to-side ratio (*Van der Perren et al., 2020*) or alkaline fibrillation conditions and higher salt concentration (*Kumari et al., 2021*) where secondary nucleation appears to be less prominent (*Buell et al., 2014*). The importance of salt for seeded polymorph replication has also been reported for the polymorph-specific seed replication between the non-ThT-sensitive ribbon polymorph and a fibril polymorph that succeeds in 150 mM KCl (with 50 mM Tris, pH 7.5, 37°C) but not in the absence of KCl (*Bousset et al., 2013*). Another approach would be to use an N-terminal-truncated version of α-Syn or to proteolytically remove the disordered C-terminus from the seeds. While it has been shown that with a larger proportion of seeds, it is possible to drive the growth by elongation instead of secondary nucleation (*Bousset et al., 2013*; *Peduzzo et al., 2020*), this may be difficult to achieve by starting from in vivo material due to limited sample amounts. Finally, the use of molecules that inhibit secondary nucleation by binding to the C-terminus of α-Syn such as the protein BRICHOS could be explored.

## Conclusion

The presented study, which focuses on the pH-dependence of fibril polymorph selection in both de novo as well as seeded fibril growth, shows the complexity of the fibril formation process, yielding many distinct polymorphs whose diversity is attributed to the absence of an evolutionary selection toward aggregation. On the other hand, it also highlights a rationale for the influence of the environment on this chameleon-like property of α-Syn. With respect to disease, the in vitro generation of a JOS-like polymorph could be obtained, and the problem of polymorph-unspecific seeding could in many experimental settings be attributed to secondary nucleation, opening an avenue to resolve this open problem in cell and animal disease studies and diagnosis.

## Methods

### Recombinant protein expression and purification of α-Syn

Recombinant N-terminally acetylated, human WT α-Syn was produced by co-expressing the pRK172 plasmid with the yeast *N*-acetyltransferase complex B (NatB) (*Johnson et al., 2010*) in *Escherichia coli* BL21(DE3*). α-Syn was expressed and purified according to the previously published protocol

(*Campioni et al., 2014*) with slight modification. In brief, a single colony from the overnight transformed plate was picked up. It was grown at 37°C in 100 ml LB media containing 100 µg/ml ampicillin and 50 µg/ml chloramphenicol. Overnight primary growth culture was added to freshly prepared 1 l LB media at a final concentration of 2.5% (vol/vol) containing 100 µg/ml ampicillin and 50 µg/ml chloramphenicol. It was further grown till O.D. reached 0.8. Protein overexpression was initiated by adding IPTG at a final concentration of 1 mM. Thereafter, cells were grown at 37°C for more than 4 hr and harvested. α-Syn was obtained from a periplasmic extract using a non-denaturing protocol. Finally, pure α-Syn was obtained by ion exchange chromatography followed by purification on a hydrophobic HiPrep phenyl FF 16/10 column (Cytiva). The pure protein was dialyzed extensively against water, lyophilized, and stored at –20°C until further use.

## Amyloid aggregation

Lyophilized α-Syn was dissolved in PBS buffer prepared from commercially available tablets (Sigma-Aldrich). The pH of the dissolved α-Syn was adjusted to pH 7.4 and then separated from aggregates on a Superdex 75 10/60 column (Cytiva) pre-equilibrated with the same buffer and pH. Monomeric α-Syn obtained from SEC was free from any preformed aggregates and its concentration was adjusted to 300 µM. 500 µl of the protein solution was incubated in a 1.5 ml Eppendorf Protein LoBind tube at gentle agitation via tumbling (~50 rpm) at 37°C for 5 days. The fibrils formed in different pH/buffer conditions were prepared by buffer exchanging the monomeric α-Syn (the elution from the Superdex 75 column) over a PD-10 desalting column (Cytiva) pre-equilibrated in the appropriate buffer and/or pH before incubation at 37°C with agitation. During every fibrillization procedure, protein concentration and incubated volume were kept constant at 300 µM and 500 µl, respectively, to ensure that different starting conditions do not affect the aggregation process. Similarly, the seeded and cross-seeded fibrils were produced by incubating 300 µM, 500 µl monomeric α-Syn in the presence of 5% (vol/vol) preformed fibrils (from a previous aggregation of 300 µM α-Syn).

## α-Syn aggregation kinetics monitored by ThT fluorescence

α-Syn aggregation kinetics in the presence and absence of preformed seeds was monitored by ThT fluorescence. Monomeric α-Syn prepared and incubated as above and at regular time intervals, 5 µl of the sample was taken and diluted to 200 µl by the same aggregation buffer. ThT was added at a final concentration of 10 µM and its fluorescence was measured by excitation at 450 nm, and emission at 480 nm with a 5 nm bandpass for both excitation and emission on a Fluoromax-4 (Jobin Yvon) spectrofluorometer. ThT fluorescence intensity at 480 nm was plotted against different time points and three independent measurements were performed for each sample. Here, only 1% of preformed aggregates were used as seeds for ThT measurement in contrast to 5% used for producing fibrils for structural studies.

## Electron microscopy grid preparation and data collection

Cu R2/2 300 mesh or Au R2/2 300 mesh grids (Quantifoils) were glow-discharged at 25 mA for 30 s and 120 s, respectively. Freshly glow-discharged grids were used in a Vitrobot Mark IV (Thermo Fisher Scientific) with its chamber set at 100% humidity and at a temperature of 22°C. Fibrils (4 µl aliquots) were applied to the grid and blotted for 8–15 s after a 30–60 s wait time, and subsequently plunge-frozen into a liquid ethane/propane mix. The grids were clipped and immediately used or stored in liquid nitrogen dewar.

Data acquisition was performed on a Titan Krios (Thermo Fisher Scientific) operating at 300 kV equipped with a Gatan Imaging Filter (GIF) with a 20 eV energy slit using Gatan's K3 direct electron detector in counting mode. Movies were collected using EPU software (Thermo Fisher Scientific) at a magnification of ×130k (see *Table 1* for physical pixel size) and a dose rate of approximately 8 e/pixel/s and total dose ca. 62–74 e/Å² (see *Table 2* for details).

## Image processing

Image processing and helical reconstruction were carried out with RELION 4.0, following the procedure for amyloid structures as described by *Scheres, 2020*. Movies were gain-corrected and RELION's own motion correction was used to correct for drift and dose-weighting and CTF estimation was done using Ctffind4.1 (*Rohou and Grigorieff, 2015*). Individual filaments were either manually selected

**Table 2.** Cryo-electron microscopy (cryo-EM) structure determination statistics.

| Polymorph | 1$_M$ | 2A | 2B | 3B | 3C | 5A | 3D (E46K) |
|---|---|---|---|---|---|---|---|
| Data collection | | | | | | | |
| Pixel size [Å] | 0.65 | 0.65 | 0.65 | 0.65 | 0.65 | 0.65 | 0.65 |
| Defocus range [μm] | –0.8 to –2.5 | 0.8 to –2.5 | 0.8 to –2.5 | –0.8 to –2.5 | –0.8 to –2.5 | –0.8 to –2.5 | –0.8 to –2.5 |
| Voltage [kV] | 300 | 300 | 300 | 300 | 300 | 300 | 300 |
| Number of frames | 40 | 40 | 40 | 40 | 40 | 40 | 40 |
| Total dose [e-/Å²] | 69 | 58.5 | 58.5 | 62 | 67 | 75 | 65 |
| Reconstruction | | | | | | | |
| Reconstruction box width [pixels] | 256 | 200 | 200 | 512 | 256 | 256 | 512 |
| Inter-box distance [Å] | 33 | 33 | 33 | 33 | 33 | 33 | 33 |
| Reconstruction pixel size [Å] | 1.3 | 1.3 | 1.3 | 0.65 | 1.3 | 1.3 | 0.65 |
| Micrographs | 1,624 | 1339 | 1339 | 7,729 | 3,127 | 1,850 | 4,666 |
| Initially extracted segments | 84,666 | 238,570 | 238,570 | 279,929 | 159,308 | 109,817 | 287,018 |
| Segments after 2D and 3D classification | 19,800 | 91,238 | 26,426 | 178,710 | 28,022 | 86,219 | 40,181 |
| 3D refinement resolution [Å] (FSC>0.143) | 3.58 | 2.99 | 3.13 | 2.64 | 3.41 | 3.40 | 2.40 |
| Final resolution [Å] (FSC >0.143) | 3.26 | 2.86 | 2.95 | 2.23 | 3.41 | 3.30 | 2.31 |
| Estimated map sharpening B-factor [Å²] | –43.8 | –73.0 | –74.9 | –51.7 | –101.7 | –87.4 | –32.8 |
| Axial symmetry | C1 | C2 | C1 | C1 | C2 | C1 | C1 |
| Helical rise [Å] | 4.79 | 4.82 | 2.39 | 2.37 | 4.77 | 2.42 | 2.41 |
| Helical twist [°] | –0.95 | –0.80 | 179.6 | 179.5 | –0.995 | 179.6 | 179.5 |
| Model composition and validation | | | | | | | |
| Non-hydrogen atoms (5 layers) | 2515 | – | – | 4550 | 4500 | 6690 | 4450 |
| Protein residues (5 layers) | 365 | – | – | 640 | 650 | 960 | 630 |
| R.m.s. deviations bond length [Å] | 0.008 | – | – | 0.005 | 0.007 | 0.006 | 0.007 |
| R.m.s. deviations bond angles [°] | 1.166 | – | – | 0.637 | 0.976 | 0.863 | 1.093 |
| MolProbity score | 1.66 | – | – | 2.01 | 1.33 | 1.64 | 1.60 |
| Clashscore | 3.71 | – | – | 7.42 | 1.62 | 4.72 | 3.57 |
| Rotamer outliers [%] | 0 | – | – | 2.22 | 1.3 | 0 | 0.68 |

*Table 2 continued on next page*

*Table 2 continued*

| Polymorph | $1_M$ | 2A | 2B | 3B | 3C | 5A | 3D (E46K) |
|---|---|---|---|---|---|---|---|
| Ramachandran plot favored [%] | 91.6 | – | – | 95.2 | 95.2 | 94.2 | 92.8 |
| Ramachandran plot allowed [%] | 81.4 | – | – | 4.8 | 4.8 | 5.8 | 7.2 |
| Ramachandran plot disallowed [%] | 0 | – | – | 0 | 0 | 0 | 0 |
| Model resolution [Å] (FSC >0.143/0.5) | 3.1/3.3 | – | – | 2.0/2.2 | 3.1/3.3 | 3.1/3.4 | 2.0/2.4 |
| PDB code | 8PK2 | – | – | 9FYP | 8PIX | 8PK4 | 8PJO |
| EMDB-ID | 17723 | 50860 | 50077 | 50888 | 17693 | 17726 | 17714 |

or selected with the Topaz implementation in RELION. The segments were extracted in a 333 Å box (except for dataset 13, *Table 1* for which a 260 Å box was used) with an inter-box distance of 33 Å and initially binned 4× to a pixel size of 2.6 Å for 2D classification. In the case of mixed polymorphs, larger box sizes were also used to help visually separate the classes and extract the crossover distances. For mixed polymorph datasets, the segments belonging to different classes were separated either by inspection of the 2D classes or by the new filament subset selection tool in RELION (version 5). For the new polymorphs ($1_M$ and 5) we then used *relion_helix_inimodel2d* to generate an initial model that could be used for the subsequent 3D refinement steps. In some datasets, the polymorphs were separated purely by 3D classification and for others purely by filament subset selection (indicated in *Table 1*). As the refinement progressed, the refined particles were re-extracted at either 1.3 Å/pixel (2× binning) or 0.65 Å/pixel (no binning) depending on the current resolution. For the deposited datasets, Bayesian polishing (*Zivanov et al., 2019*) and CTF refinement (*Zivanov et al., 2020*)

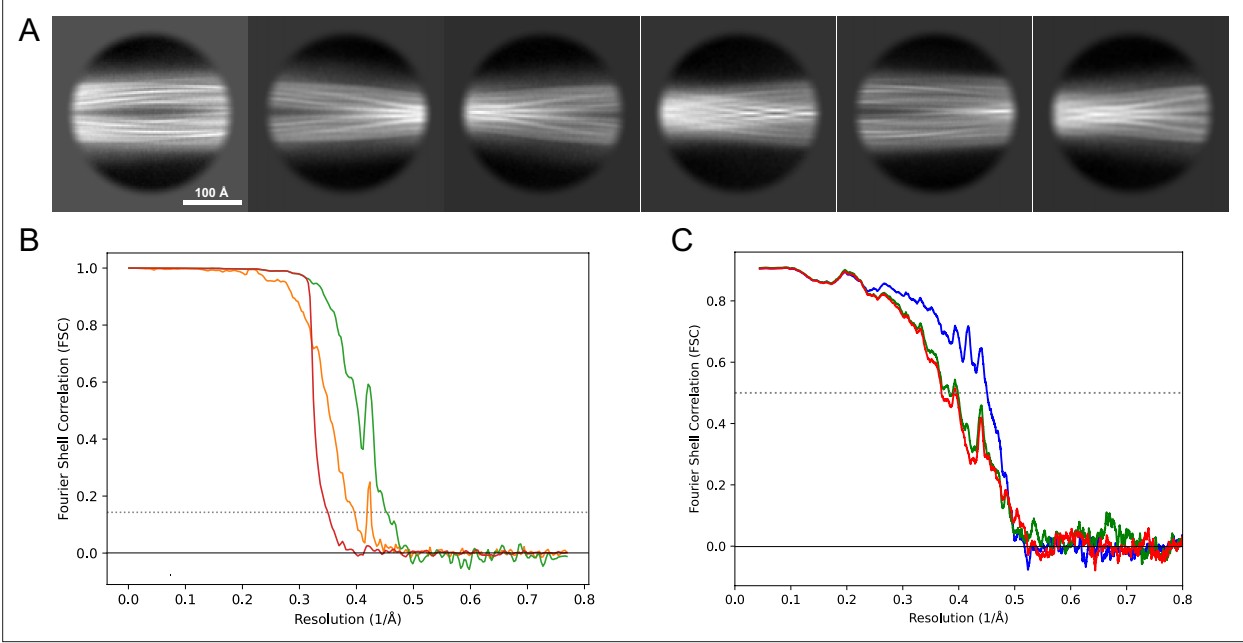

**Figure 11.** 2D classes and half-map and model-map FSC curves for dataset 1, Type 3B. (**A**) Representative 2D classes of the segments that were used for the 3D reconstruction. (**B**) The FSC curves produced during postprocessing in RELION with red showing the plot for the phase randomized, orange the unmasked, and green the masked maps. (**C**) The model-map FSC curves produced in PHENIX. The blue curve is for the deposited coordinates and full postprocessed map against which it had been fit by real-space refinement in PHENIX. Coordinates were similarly generated by refining against the first half-map and then compared to the same half-map (green) or the second half-map (red).

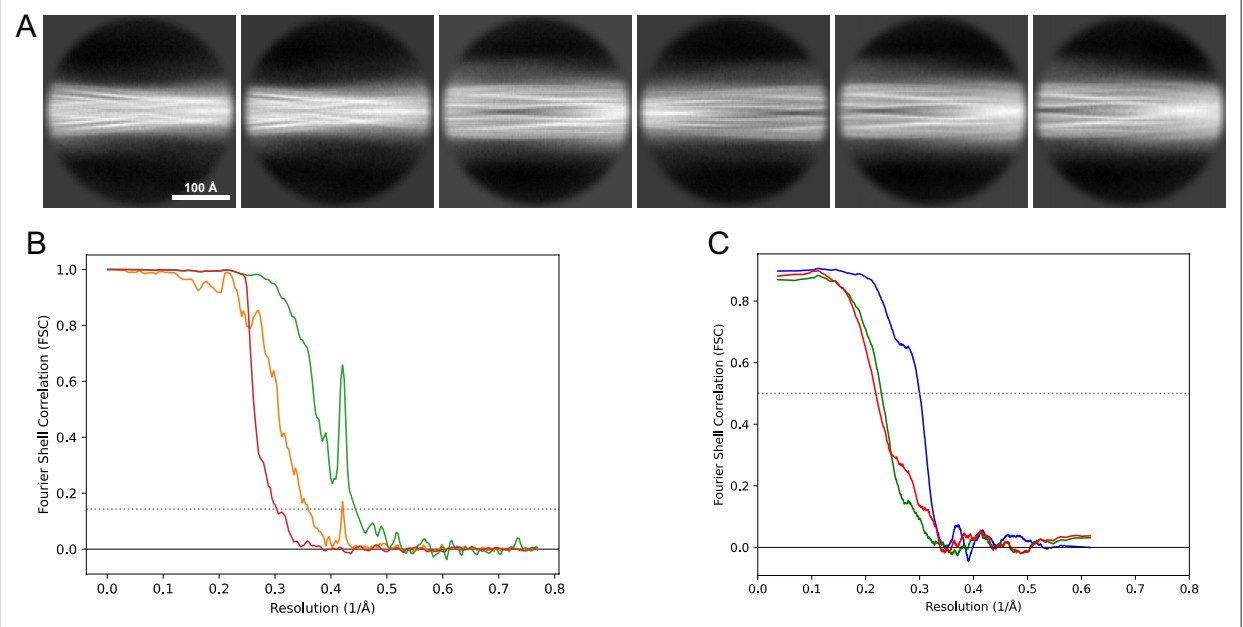

**Figure 12.** 2D classes and half-map and model-map FSC curves for dataset 3, Type 3C. (**A**) Representative 2D classes of the segments that were used for the 3D reconstruction. (**B**) The FSC curves produced during postprocessing in RELION with red showing the plot for the phase randomized, orange the unmasked, and green the masked maps. (**C**) The model-map FSC curves produced in PHENIX. The blue curve is for the deposited coordinates and full postprocessed map against which it had been fit by real-space refinement in PHENIX. Coordinates were similarly generated by refining against the first half-map and then compared to the same half-map (green) or the second half-map (red).

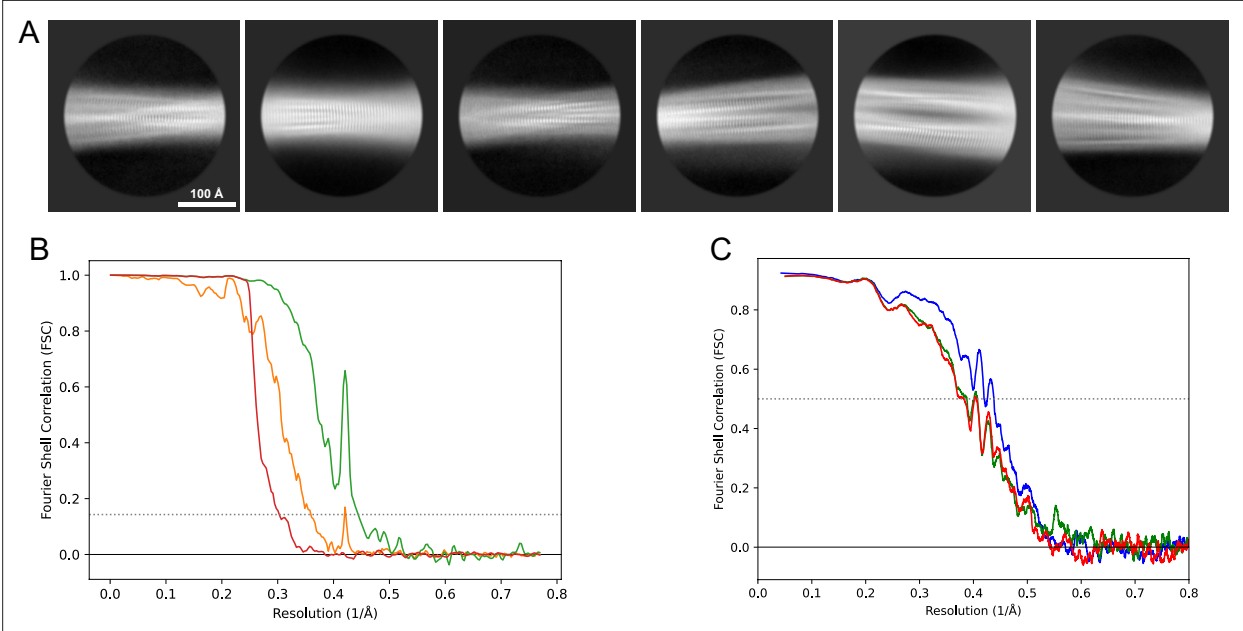

**Figure 13.** 2D classes and half-map and model-map FSC curves for dataset 5, Type 3D. (**A**) Representative 2D classes of the segments that were used for the 3D reconstruction. (**B**) The FSC curves produced during postprocessing in RELION with red showing the plot for the phase randomized, orange the unmasked, and green the masked maps. (**C**) The model-map FSC curves produced in PHENIX. The blue curve is for the deposited coordinates and full postprocessed map against which it had been fit by real-space refinement in PHENIX. Coordinates were similarly generated by refining against the first half-map and then compared to the same half-map (green) or the second half-map (red).

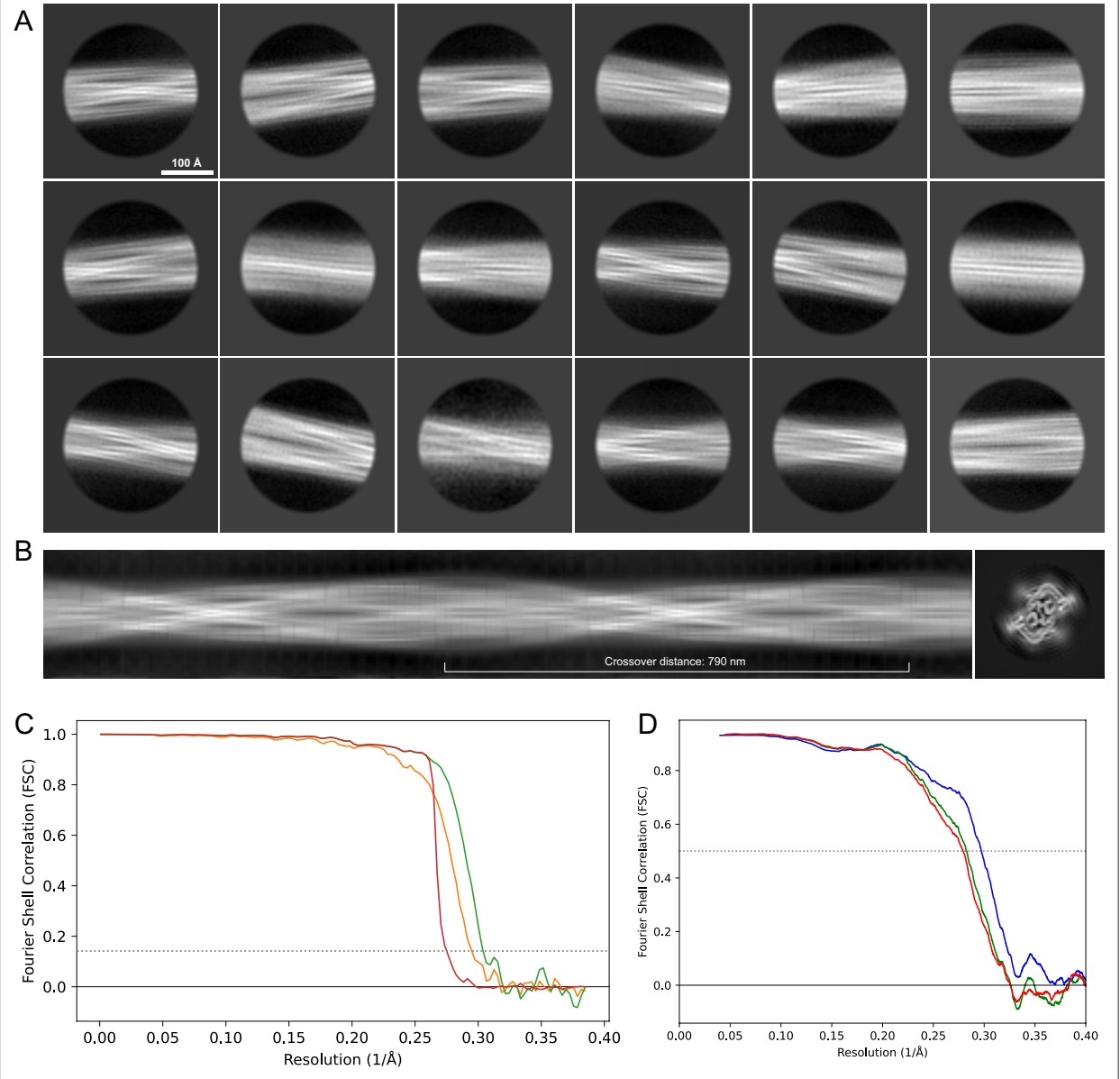

**Figure 14.** 2D classes and half-map and model-map FSC curves for dataset 8, Type 5. (**A**) The 2D classes of the segments that were used to produce the initial model in relion_helix_inimodel2d. (**B**) The output of relion_helix_inimodel2d shown as the summed 2D classes and a Z-projection of the reconstructed 3D model used as input for refinement in the 3D refinement. (**C**) The FSC curves produced during postprocessing in RELION with red showing the plot for the phase randomized, orange the unmasked, and green the masked 2D classes and half-map FSC curves for dataset 13, Type 2A.maps. (**D**) The model-map FSC curves produced in PHENIX. The blue curve is for the deposited coordinates and full postprocessed map against which it had been fit by real-space refinement in PHENIX. Coordinates were similarly generated by refining against the first half-map and then compared to the same half-map (green) or the second half-map (red).

were performed to improve the resolution of the reconstructions. The deposited reconstructions were sharpened using standard postprocessing procedures in RELION. For the data for which maps have been deposited in the EMDB (*Table 2*), representative 2D classes and the postprocess-generated FSC curves are shown in *Figure 11*, *Figure 12*, *Figure 13*, *Figure 14*, *Figure 15*, *Figure 16*, *Figure 17*.

## Model building and refinement

Models were built into the RELION postprocessed maps with ModelAngelo (*Jamali et al., 2023*), the output of which was manually adjusted in COOT (*Emsley and Cowtan, 2004*) and further adjusted by real-space refinement as a nine-layer fibril in ISOLDE (*Croll, 2018*) with symmetry and secondary

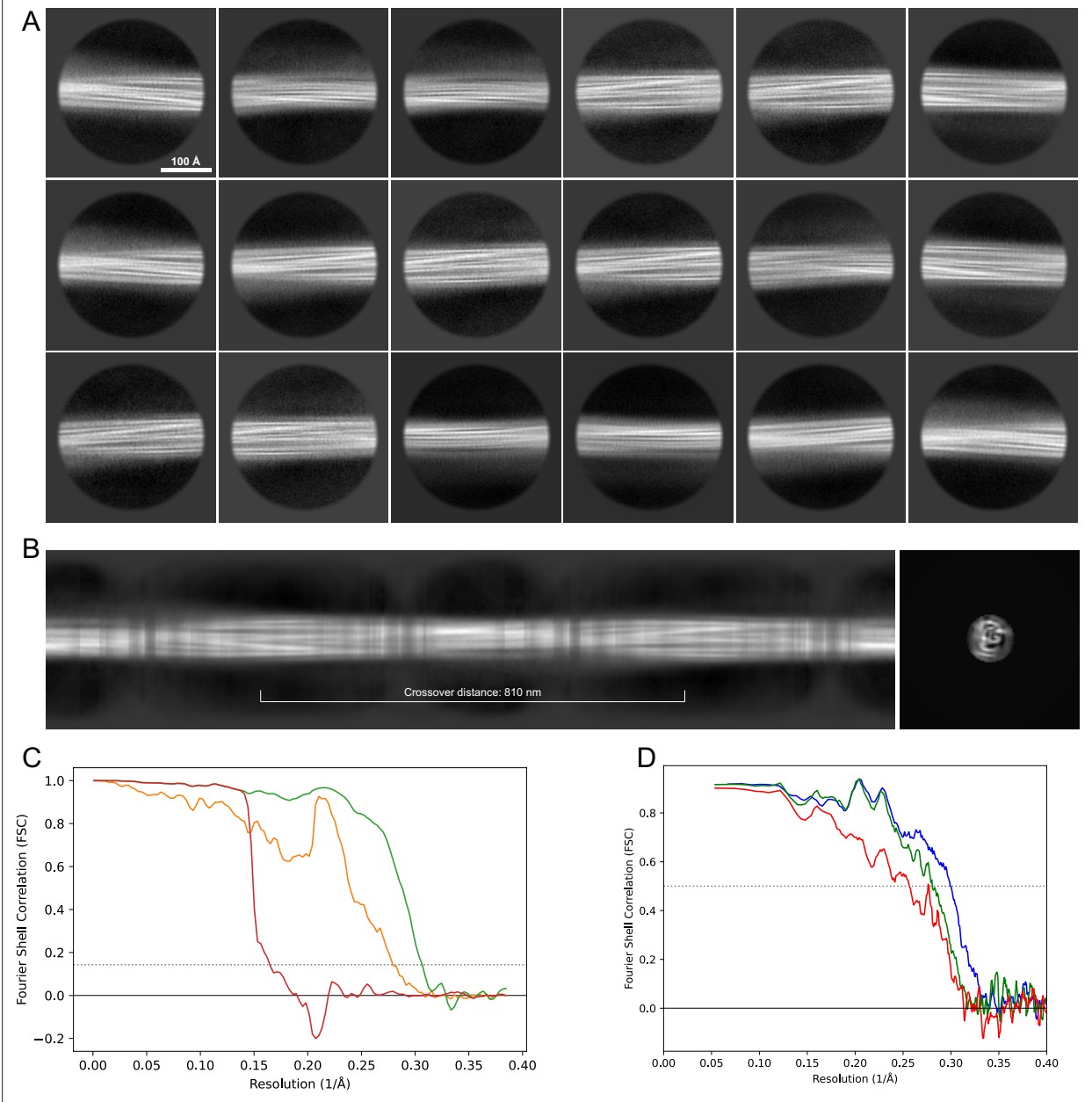

**Figure 15.** 2D classes and half-map and model-map FSC curves for dataset 9, Type $1_M$. (**A**) The 2D classes of the segments that were used to produce the initial model in relion_helix_inimodel2d. (**B**) The output of relion_helix_inimodel2d shown as the summed 2D classes and a Z-projection of the reconstructed 3D model used as input for refinement in the 3D refinement. (**C**) The FSC curves produced during postprocessing in RELION with red showing the plot for the phase randomized, orange the unmasked, and green the masked maps. (**D**) The model-map FSC curves produced in PHENIX. The blue curve is for the deposited coordinates and full postprocessed map against which it had been fit by real-space refinement in PHENIX. Coordinates were similarly generated by refining against the first half-map and then compared to the same half-map (green) or the second half-map (red).

structure restraints. The outer four layers, which often diverge slightly in structure due to their placement at the edges of the model, were removed and the central five layers further refined in PHENIX (**Liebschner et al., 2019**) to obtain reasonable b-factors before deposition in the PDB. Figures were prepared with CCP4MG (**McNicholas et al., 2011**) and UCSF Chimera (**Pettersen et al., 2004**). For the data for which coordinates have been deposited in the PDB (**Table 2**), model-map FSC curves generated with the validation tools in PHENIX (**Afonine et al., 2018**) are shown in **Figure 11**, **Figure 12**,

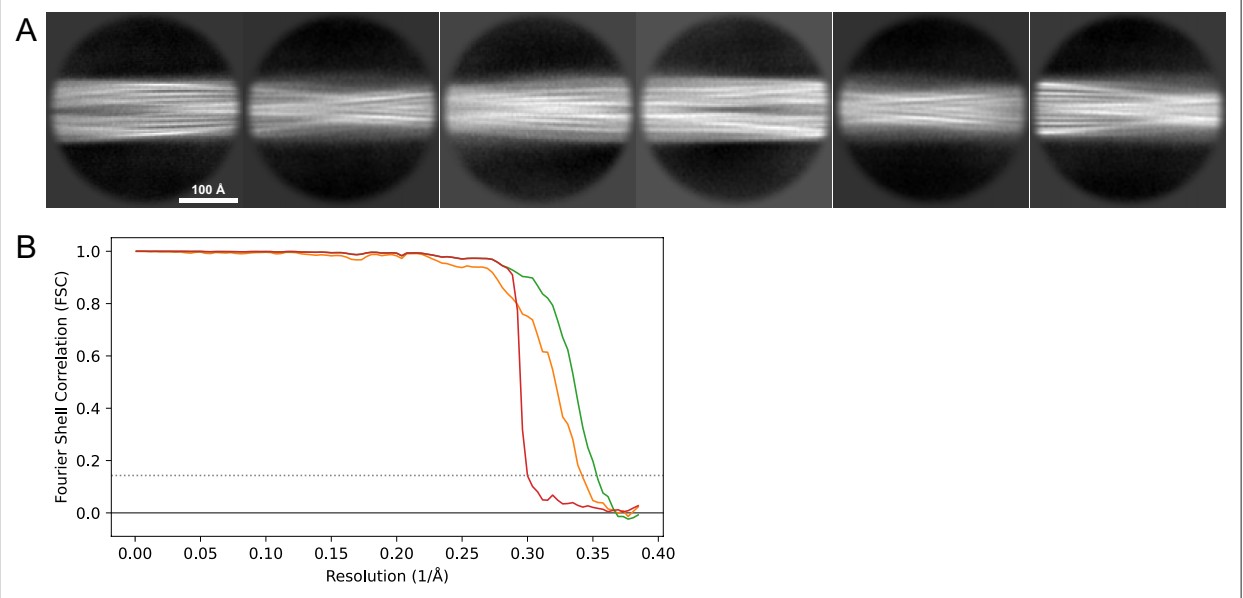

**Figure 16.** 2D classes and half-map FSC curves for dataset 13, Type 2A. (**A**) Representative 2D classes of the segments that were used for the 3D reconstruction. (**B**) The FSC curves produced during postprocessing in RELION with red showing the plot for the phase randomized, orange the unmasked, and green the masked maps.

*Figure 13*, *Figure 14*, *Figure 15*. For the model-map analyses, the maps were first masked to contain just the five-layer PDB file.

## Limited proteolysis mass spectrometry

LiP-MS experiments were conducted as previously described (*Schopper et al., 2017*; *Malinovska et al., 2023*). Briefly, Proteinase K (PK) was added at a 1:100 enzyme-to-substrate ratio to 2 μg of α-Syn in 100 μl of LiP-buffer. Four independent PK digestions (technical replicates) were performed for each condition at 37°C for 5 min. Subsequently, PK activity was halted by incubating the samples at 99°C for 5 min. After the PK heat-inactivation step, all the samples were incubated

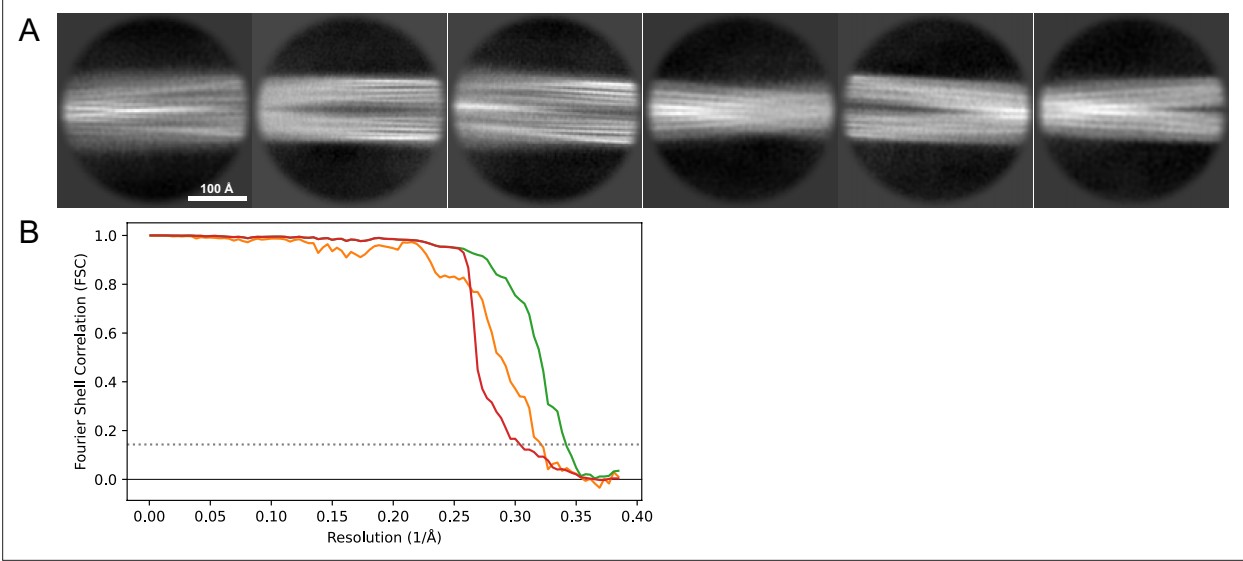

**Figure 17.** 2D classes and half-map FSC curves for dataset 13, Type 2B. (**A**) Representative 2D classes of the segments that were used for the 3D reconstruction. (**B**) The FSC curves produced during postprocessing in RELION with red showing the plot for the phase randomized, orange the unmasked, and green the masked maps.

for 5 min on ice. Sodium deoxycholate (DOC, 10%) was added to each sample to achieve a final DOC concentration of 5%. Next, the samples were reduced with tris(2-carboxyethyl)phosphine hydrochloride at a final concentration of 5 mM at 37°C for 40 min with shaking at 800 rpm in a thermomixer (Eppendorf). Alkylation was performed with iodoacetamide at a final concentration of 40 mM in the dark for 30 min at room temperature. To decrease the DOC concentration to 1%, the samples were diluted with 10 mM ammonium bicarbonate. Next, sequencing-grade porcine trypsin (Promega) and Lysyl endopeptidase LysC (Wako Chemicals) were added to the samples at an enzyme-to-substrate ratio of 1:100. The samples were digested overnight in a thermomixer at 37°C with continuous agitation at 800 rpm. After overnight incubation, the digestion was stopped by adding 100% formic acid (FA) (Carl Roth GmbH) to achieve a final concentration of 2%. Finally, precipitated DOC was removed by three cycles of 20 min centrifugation at 21,000 rcf. The samples were loaded onto Sep-Pak C18 cartridges (Waters), desalted, and subsequently eluted with 80% acetonitrile, 0.1% FA. The samples were analyzed by Orbitrap Eclipse Tribrid Mass Spectrometer (Thermo Fisher Scientific) in DIA mode (*Gillet et al., 2012*; *Cappelletti et al., 2021*). The raw MS data were searched with Spectronaut (Biognosys AG, version 15) and statistical data analysis was performed in R version 4.3.1. A two-tailed t-test was used to calculate p-values. Scores were designated to each detected peptide, calculated as a product of the $-\log_{10}$(p-value) and $|\log_2$(fold change)$|$. Subsequently, we calculated the mean value of this score for each amino acid based on overlapping peptides.

## Acknowledgements

We would like to thank the staff at the ScopeM ETHZ for assistance with the cryo-EM data collection analysis. This research was supported by the Swiss National Science Foundation (grant number 205320_182800).

## Additional information

### Funding

| Funder | Grant reference number | Author |
|---|---|---|
| Swiss National Science Foundation | 205320_182800 | Roland Riek |

The funders had no role in study design, data collection and interpretation, or the decision to submit the work for publication.

### Author contributions

Lukas Frey, Conceptualization, Formal analysis, Investigation, Visualization, Writing – review and editing; Dhiman Ghosh, Investigation, Methodology; Bilal M Qureshi, Resources, Data curation; David Rhyner, Formal analysis, Visualization, Writing – review and editing; Ricardo Guerrero-Ferreira, Supervision; Aditya Pokharna, Paola Picotti, Investigation; Witek Kwiatkowski, Conceptualization, Data curation, Formal analysis, Validation, Investigation, Writing - original draft, Writing – review and editing; Tetiana Serdiuk, Investigation, Writing - original draft; Roland Riek, Conceptualization, Supervision, Writing – review and editing; Jason Greenwald, Formal analysis, Supervision, Writing - original draft, Writing – review and editing

### Author ORCIDs

Ricardo Guerrero-Ferreira  https://orcid.org/0000-0002-3664-8277
Jason Greenwald  https://orcid.org/0000-0002-9448-4821

Reviewer #2 (Public Review): https://doi.org/10.7554/eLife.93562.4.sa1
Reviewer #3 (Public Review): https://doi.org/10.7554/eLife.93562.4.sa2
Author response https://doi.org/10.7554/eLife.93562.4.sa3

## Additional files

### Supplementary files
• MDAR checklist

### Data availability

The Cryo-EM maps have been deposited in the EMDB under the accession codes: 17723, 50860, 50077, 50888, 17693, 17726 and 17714. The models have been deposited in the PDB under the accession codes: 8PK2, 9FYP, 8PIX, 8PK4 and 8PJO.

The following datasets were generated:

| Author(s) | Year | Dataset title | Dataset URL | Database and Identifier |
|---|---|---|---|---|
| Frey L, Qureshi BM, Kwiatkowski W, Rhyner D, Greenwald J, Riek R | 2024 | Cryo EM structure of the type 3C polymorph of alpha-synuclein at low pH | https://www.rcsb.org/structure/8PIX | RCSB Protein Data Bank, 8PIX |
| Frey L, Qureshi BM, Kwiatkowski W, Rhyner D, Greenwald J, Riek R | 2024 | Cryo EM structure of the type 3D polymorph of alpha-synuclein E46K mutant at low pH | https://www.rcsb.org/structure/8PJO | RCSB Protein Data Bank, 8PJO |
| Frey L, Qureshi BM, Kwiatkowski W, Rhyner D, Greenwald J, Riek R | 2024 | Cryo EM structure of the type 1m polymorph of alpha-synuclein | https://www.rcsb.org/structure/8PK2 | RCSB Protein Data Bank, 8PK2 |
| Frey L, Qureshi BM, Kwiatkowski W, Rhyner D, Greenwald J, Riek R | 2024 | Cryo EM structure of the type 5A polymorph of alpha-synuclein | https://www.rcsb.org/structure/8PK4 | RCSB Protein Data Bank, 8PK4 |
| Frey L, Qureshi BM, Kwiatkowski W, Rhyner D, Greenwald J, Riek R | 2024 | Cryo EM structure of the type 3B polymorph of alpha-synuclein at low pH | https://www.rcsb.org/structure/9FYP | RCSB Protein Data Bank, 9FYP |
| Frey L, Qureshi BM, Kwiatkowski W, Rhyner D, Greenwald J, Riek R | 2024 | Cryo EM structure of the type 3C polymorph of alpha-synuclein at low pH | https://www.ebi.ac.uk/emdb/EMD-17693 | Electron Microscopy Data Bank, 17693 |
| Frey L, Qureshi BM, Kwiatkowski W, Rhyner D, Greenwald J, Riek R | 2024 | Cryo EM structure of the type 3D polymorph of alpha-synuclein E46K mutant at low pH | https://www.ebi.ac.uk/emdb/EMD-17714 | EMDB, 17714 |
| Frey L, Qureshi BM, Kwiatkowski W, Rhyner D, Greenwald J, Riek R | 2024 | Cryo EM structure of the type 1m polymorph of alpha-synuclein | https://www.ebi.ac.uk/emdb/EMD-17723 | Electron Microscopy Data Bank, 17723 |
| Frey L, Qureshi BM, Kwiatkowski W, Rhyner D, Greenwald J, Riek R | 2024 | Cryo EM structure of the type 5A polymorph of alpha-synuclein | https://www.ebi.ac.uk/emdb/EMD-17726 | Electron Microscopy Data Bank, 17726 |
| Frey L, Qureshi BM, Kwiatkowski W, Rhyner D, Greenwald J, Riek R | 2024 | Cryo EM structure of the type 3B polymorph of alpha-synuclein at low pH | https://www.ebi.ac.uk/emdb/EMD-50888 | Electron Microscopy Data Bank, 50888 |
| Frey L, Qureshi BM, Kwiatkowski W, Rhyner D, Greenwald J, Riek R | 2024 | Cryo EM map of the type 2A polymorph of alpha-synuclein at pH 7.0 | https://www.ebi.ac.uk/emdb/EMD-50860 | Electron Microscopy Data Bank, 50860 |
| Frey L, Qureshi BM, Kwiatkowski W, Rhyner D, Greenwald J, Riek R | 2024 | Cryo EM map of the type 2B polymorph of alpha-synuclein at pH 7.0 | https://www.ebi.ac.uk/emdb/EMD-50077 | Electron Microscopy Data Bank, 50077 |

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
