## [Editor Report · eLife assessment]

This study presents **important** findings on the different polymorphs of alpha-synuclein filaments that form at various pH's during in vitro assembly reactions with purified recombinant protein. Of particular note is the discovery of two new polymorphs (1M and 5A) that form in PBS buffer at pH 7. The strength of the evidence presented is **convincing**. The work will be of interest to biochemists and biophysicists working on protein aggregation and amyloids.

---

## [Referee Report · Reviewer #2 (Public Review)]

Summary:

This is an exciting paper that explores the in vitro assembly of recombinant alpha-synuclein into amyloid filaments. The authors changed the pH and the composition of the assembly buffers, as well as the presence of different types of seeds, and analysed the resulting structures by cryo-EM.

Strengths:

By doing experiments at different pHs, the authors found that so-called type 2 and type-3 polymorphs form in a pH dependent manner. In addition, they find that type-1 filaments form in the presence of phosphate ions. One of their in vitro assembled type-1 polymorphs is similar to the alpha-synuclein filaments that were extracted from the brain of an individual with juvenile-onset synucleinopathy (JOS). They hypothesize that additional densities in a similar place as additional densities in the JOS fold correspond to phosphate ions.

Comments on the revised version:

This is OK now. I thank the authors for their constructive engagement with my comments.

---

## [Referee Report · Reviewer #3 (Public Review)]

Summary

The high heterogeneity nature of α-synuclein (α-syn) fibrils posed significant challenges in structural reconstruction of the ex vivo conformation. A deeper understanding of the factors influencing the formation of various α-syn polymorphs remains elusive. The manuscript by Frey et al. provides a comprehensive exploration of how pH variations (ranging from 5.8 to 7.4) affect the selection of α-syn polymorphs (specifically, Type1, 2 and 3) in vitro by using cryo-electron microscopy (cryo-EM) and helical reconstruction techniques. Crucially, the authors identify two novel polymorphs at pH 7.0 in PBS. These polymorphs bear resemblance to the structure of patient-derived juvenile-onset synucleinopathy (JOS) polymorph and diseased tissue amplified α-syn fibrils. The revised manuscript more strongly supports the notion that seeding is a non-polymorph-specific in the context of secondary nucleation-dominated aggregation, underscoring the irreplaceable role of pH in polymorph formation.

Strengths

This study systematically investigates the effects of environmental conditions and seeding on the structure of α-syn fibrils. It emphasizes the significant influence of environmental factors, especially pH, in determining the selection of α-syn polymorphs. The high-resolution structures obtained through cryo-EM enable a clear characterization of the composition and proportion of each polymorph in the sample. Collectively, this work provides a strong support for the pronounced sensitivity of α-syn fibril structures to the environmental conditions and systematically categorizes previously reported α-syn fibril structures. Furthermore, the identification of JOS-like polymorph also demonstrates the possibility of in vitro reconstruction of brain-derived α-syn fibril structures.

Weaknesses

All my previous concerns have been resolved to my satisfaction.

---

## [Author Response]

The following is the authors’ response to the previous reviews.

Revisions Round 1

**Reviewer #1**

We thank the reviewer for their careful reading of our manuscript and have taken all of their grammatical corrections into account.

**Reviewer #2 (Public Review):**
Weaknesses:The paper contains multiple instances of non-scientific language, as indicated below. It would also benefit from additional details on the cryo-EM structure determination in the Methods and inclusion of commonly accepted requirements for cryo-EM structures, like examples of 2D class averages, raw micrographs, and FSC curves (between half-maps as well as between rigid-body fitted (or refined) atomic models of the different polymorphs and their corresponding maps). In addition, cryo-EM maps for the control experiments F1 and F2 should be presented in Figure 9.

We tried to correct the non-scientific language and have included the suggested data on the Cryo-EM analyses including new Figures 11-17. We did not collect data on the sample used for the seeds in the cross seeding experiments because we had already confirmed in multiple datasets that the conditions in F1 and F2 reproducibly produce fibrils of Type 1 and Type 3, respectively. We have now analyzed cryo-EM data for 6 more samples at pH 7.0 and found that several kinds of polymorphs (Types 1A, 1M, 2A, 2B and 5) are accessible at this pH, however the Type 3 polymorphs are not formed at pH 7.0 under the conditions that we used for aggregation.

**Reviewer #2 (Recommendations For The Authors):**
- remove unscientific language: "it seems that there are about as many unique atomic-resolution structures of these aggregates as there are publications describing them"

We have rephrased this sentence.

- for same reason, remove "Obviously, "

Done

- What does this mean? “polymorph-unspecific”

Rephrased as non-polymorph-specific

- What does this mean? "shallow amyloid energy hypersurface"

By “shallow hypersurface” we mean that the minimum of the multi-dimensional function that describes the energy of the amyloid is not so deep that subtle changes to the environment will not favor another fold/energy minimum. We have left the sentence because while it may not be perfect, it is concise and seems to get the point across.

- "The results also confirm the possibility of producing disease-relevant structure in vitro." -> This is incorrect as no disease-relevant structure was replicated in this work. Use another word like “suggest”.

We have changed to “suggest” as suggested.

- Remove "historically"

Done

- Rephrase “It has long been understood that all amyloids contain a common structural scaffold”

Changed to “It has long been established that all amyloids contain a common structural scaffold..”

- "Amyloid polymorphs whose differences lie in both their tertiary structure (the arrangement of the beta-strands) and the quaternary structure (protofilamentprotofilament assembly) have been found to display distinct biological activities [8]" -> I don't think this is true, different biological activities of amyloids have never been linked to their distinct structures.

We have added 5 new references (8-12) to support this sentence.

- Reference 10 is a comment on reference 9; it should be removed. Instead, as for alpha-synuclein, all papers describing the tau structures should be included.

We have removed the reference, but feel that the addition of all Tau structure references is not merited in this manuscript since we are not comparing them.

- Rephrase: "is not always 100% faithful"

Removed “100%”

- What is pseudo-C2 symmetry? Do the authors mean pseudo 2_1 symmetry (ie a 2-start helical symmetry)?

Thank for pointing this out. We did indeed mean pseudo 21 helical symmetry.

- Re-phrase: "alpha-Syn's chameleon-like behavior"

We have removed this phrase.

- "In the case of alpha-Syn, the secondary nucleation mechanism is based on the interaction of the positively charged N-terminal region of monomeric alpha-Syn and the disordered, negatively charged C-terminal region of the alpha-Syn amyloid fibrils [54]" -> I would say the mechanisms of secondary nucleation are not that well understood yet, so one may want to tune this down a bit.

We have changed this to “mechanism has been proposed to be”

- The paragraphs describing experiments by others are better suited for a Discussion rather than a Results section. Perhaps re-organize this part?

We have left the text intact as we are using a Results and Discussion format.

- A lot of information about Image processing seems to be missing: what steps were performed after initial model generation?

We have added more details in the methods section on the EM data processing and model analysis.

- Figure 1: Where is Type 4 on the pH scale?

We have adjusted the Fig 1 legend to clarify that pH scale is only applicable to the structures presented in this manuscript.

- Figure 2: This might be better incorporated as a subpanel of Figure 1.

We agree that this figure is somewhat of a loner on its own and we only added it in order to avoid confusion with the somewhat inconsistent naming scheme used for the Type 1B structure. However, we prefer to leave it as a separate figure so that it does not get dilute the impact of figure 1.

- Figure 3: What is the extra density at the bottom of Type 3B from pH 5.8 samples 1 and 2. pH 5.8 + 50mM NaCl (but not pH 5.8 + 100 mM NaCl)? Could this be an indication of a local minimum and the pH 5.8 + 100 mM NaCl structure is correct? Or is this a real difference between 0/50mM NaCl and 100 mM NaCl?

We did not see the extra density to which the reviewer is referring, however the images used in this panel are the based on the output of 3D-classification which is more likely to produce more artifacts than a 3D refinement. With this in mind, we did not see any significant differences in the refined structures and therefore only deposited the better quality map and model for each of the polymorph types.

- Figure 3: To what extent is Type 3B of pH 6.5 still a mixture of different types? The density looks poor. In general, in the absence of more details about the cryo-EM maps, it is hard to assess the quality of the structures presented.

In order to improve the quality of the images in this panel, a more complete separation of the particles from each polymorph was achieved via the filament subset selection tool in RELION 5. In each case, an unbiased could be created from the 2D classes via the relion_helix_inimodel2D program, further supporting the coexistence of 4 polymorphs in the pH 6.5 sample. The particles were individually refined to produce the respective maps that are now used in this figure.

- Many references are incorrect, containing "Preprint at (20xx)" statements.

This has been corrected.

**Reviewer #3 (Public Review):**
Weaknesses:(1) The authors reveal that both Type 1 monofilament fibril polymorph (reminiscent of JOSlike polymorph) and Type 5 polymorph (akin to tissue-amplified-like polymorph) can both form under the same condition. Additionally, this condition also fosters the formation of flat ribbon-like fibril across different batches. Notably, at pH 5.8, variations in experimental groups yield disparate abundance ratios between polymorph 3B and 3C, indicating a degree of instability in fibrillar formation. The variability would potentially pose challenges for replicability in subsequent research. In light of these situations, I propose the following recommendations:(a) An explicit elucidation of the factors contributing to these divergent outcomes under similar experimental conditions is warranted. This should include an exploration of whether variations in purified protein batches are contributing factors to the observed heterogeneity.

We are in complete agreement that understanding the factors that lead to polymorph variability is of utmost importance (and was the impetus for the manuscript itself). However the number of variables to explore is overwhelming and we will continue to investigate this in our future research. Regarding the variability between batches of purified protein, we also think that this could be a factor in the polymorph variability observed for otherwise “identical” aggregation conditions, particularly at pH 7 where the largest variety of polymorphs have been observed. However, even variation between identical replicates (samples created from the same protein solution and simply aggregated simultaneously in separate tubes) can lead to different outcomes (see datasets 15 and 16 in the revised Table 1) suggesting that there are stochastic processes that can determine the outcome of an individual aggregation experiment. While our data still indicates that Type 1,2 and 3 polymorphs are strongly selected by pH, the selection between interface variants 3B vs. 3C and 2A vs. 2B might also be affected by protein purity. Our standard purification protocol produces a single band by coomassie-stained SDS-PAGE however minor truncations and other impurities below a few percent would go undetected and, given the proposed roles of the N and C-termini in secondary nucleation, could have a large effect on polymorph selection and seeding. In line with the reviewer’s comments we now include a batch number for each EM dataset. While no new conclusions can be drawn from the inclusion of this additional data, we feel that it is important to acknowledge the possible role of batch to batch variability.

(b) To enhance the robustness of the conclusions, additional replicates of the experiments under the same condition should be conducted, ideally a minimum of three times.

The pH 5.8 conditions that yield Type 3 fibrils has already been repeated several times in the original manuscript. Since the pH 7.4 conditions produce the most common a-Syn polymorph (Type 1A) and were produced twice in this manuscript (once as an unseeded and once as a cross-seeded fibrilization) we decided to focus on the intermediate condition where the most variability had been seen (pH 7.0). The revised table 1 now has 6 new datasets (11-16) representing 6 independent aggregations at pH 7.0 starting from two different protein purification batches. The results is that we now produce the type 2A/B polymorphs in three samples and in two of these samples we once again observed the type 1M polymorph. The other samples produced Type 1A or non-twisted fibrils.

(c) Further investigation into whether different polymorphs formed under the same buffer condition could lead to distinct toxicological and pathology effects would be a valuable addition to the study.

The correlation of toxicity with structure would in principle be interesting. However the Type 1 and Type 3 polymorphs formed at pH 5.8 and 7.4 are not likely to be biologically relevant. The pH 7 polymorphs (Type 5 and 1M) would be more interesting because they form under the same conditions and might be related to some disease relevant structures. Still, it is rare that a single polymorph appears at 7.0 (the Type 5 represented only 10-20% of the fibrils in the sample and the Type 1M also had unidentified double-filament fibrils in the sample). We plan to pursue this line of research and hope to include it in a future publication.

(2) The cross-seeding study presented in the manuscript demonstrates the pivotal role of pH conditions in dictating conformation. However, an intriguing aspect that emerges is the potential role of seed concentration in determining the resultant product structure. This raises a critical question: at what specific seed concentration does the determining factor for polymorph selection shift from pH condition to seed concentration? A methodological robust approach to address this should be conducted through a series of experiments across a range of seed concentrations. Such an approach could delineate a clear boundary at which seed concentration begins to predominantly dictate the conformation, as opposed to pH conditions. Incorporating this aspect into the study would not only clarify the interplay between seed concentration and pH conditions, but also add a fascinating dimension to the understanding of polymorph selection mechanisms.

A more complete analysis of the mechanisms of aggregation, including the effect of seed concentration and the resulting polymorph specificity of the process, are all very important for our understanding of the aggregation pathways of alpha-synuclein and are currently the topic of ongoing investigations in our lab.

Furthermore, the study prompts additional queries regarding the behavior of cross-seeding production under the same pH conditions when employing seeds of distinct conformation. Evidence from various studies, such as those involving E46K and G51D cross-seeding, suggests that seed structure plays a crucial role in dictating polymorph selection. A key question is whether these products consistently mirror the structure of their respective seeds.

We thank the reviewer for reminding us to cite these studies as a clear example of polymorph selection by cross-seeding. Unfortunately, it is not 100% clear from the G51D cross seeding manuscript (https://doi.org/10.1038/s41467-021-26433-2) what conditions were used in the cross-seeding since different conditions were used for the seedless wild-type and mutant aggregations… however it appears that the wildtype without seeds was Tris pH 7.5 (although at 37C the pH could have dropped to 7ish) and the cross-seeded wild-type was in Phosphate buffer at pH 7.0. In the E46K cross-seeding manuscript, it appears that pH 7.5 Tris was used for all fibrilizations (https://doi.org/10.1073/pnas.2012435118). In any event, both results point to the fact that at pH 7.0-7.5 under low-seed conditions (0.5%) the Type 4 polymorph can propagate in a seed specific manner.

(3) In the Results section of "The buffer environment can dictate polymorph during seeded nucleation", the authors reference previous cell biological and biochemical assays to support the polymorph-specific seeding of MSA and PD patients under the same buffer conditions. This discussion is juxtaposed with recent research that compares the in vivo biological activities of hPFF, ampLB as well as LB, particularly in terms of seeding activity and pathology. Notably, this research suggests that ampLB, rather than hPFF, can accurately model the key aspects of Lewy Body Diseases (LBD) (refer to: https://doi.org/10.1038/s41467-023-42705-5). The critical issue here is the need to reconcile the phenomena observed in vitro with those in in-vivo or in-cell models. Given the low seed concentration reported in these studies, it is imperative for the authors to provide a more detailed explanation as to why the possible similar conformation could lead to divergent pathologies, including differences in cell-type preference and seeding capability.

We thank the reviewer for bring this recent report to our attention. The findings that ampLB and hPFF have different PK digestion patterns and that only the former is able to model key aspects of Lewy Body disease are in support of the seed-specific nature of some types of alpha-synuclein aggregation. We have added this to the discussion regarding the significant role that seed type and seed conditions likely play in polymorph selection.

(4) In the Method section of "Image processing", the authors describe the helical reconstruction procedure, without mentioning much detail about the 3D reconstruction and refinement process. For the benefit of reproducibility and to facilitate a deeper understanding among readers, the authors should enrich this part to include more comprehensive information, akin to the level of detail found in similar studies (refer to: https://doi.org/10.1038/nature23002).

As also suggested by reviewer #2, we have now added more comprehensive information on the 3D reconstruction and refinement process.

(5) The abbreviation of amino acids should be unified. In the Results section "On the structural heterogeneity of Type 1 polymorphs", the amino acids are denoted using three-letter abbreviation. Conversely, in the same section under "On the structural heterogeneity of Type 2 and 3 structures", amino acids are abbreviated using the one-letter format. For clarity and consistency, it is essential that a standardized format for amino acid abbreviations be adopted throughout the manuscript.

That makes perfect sense and had been corrected.

**Reviewing Editor:**
After discussion among the reviewers, it was decided that point 2 in Reviewer #3's Public Review (about the experiments with different concentrations of seeds) would probably lie outside the scope of a reasonable revision for this work.

We agree as stated above and will continue to work on this important point.

Revisions Round 2

**Reviewer #2 (Public Review):**
I do worry that the FSC values of model-vs-map appear to be higher than expected from the corresponding FSCs between the half-maps (e.g. see Fig 13). The implication of this observation is that the atomic models may have been overfitted in the maps, which would have led to a deterioration of their geometry. A table with rmsd on bond lengths, angles, etc would probably show this. In addition, to check for overfitting, the atomic model for each data set could be refined in one of the half-maps, and then that same model could be used to calculate 2 FSC model-vs-map curves: one against the half-map it was refined in and one against the other half-map. Deviations between these two curves are an indication of overfitting.

Thank you for the recommendations for model validation. We have added the suggested statistics to Table 2 and performed the suggested model fitting to one of the half-maps and plotted 3 FSC model-vs-map curves: one for each half-map versus the model fit against only one half map and one for the model fit against the full map. We feel that the degree of overfitting is reasonable and does not significantly impact the quality of the models.

In addition, the sudden drop in the FSC curves in Figure 16 shows that something unexpected has happened to this refinement. Are the authors sure that only the procedures outlined in the Methods were used to create these curves? The unexpected nature of the FSC curve for this type (2A) raises doubts about the correctness of the reconstruction.

We thank the reviewer for the attention to detail. We should have caught this mistake. It turns out that in the last round of 3D refinement, the two half-maps become shifted with respect to each other in the z direction. We realigned the two maps using Chimera and then re-ran the postprocessing. The new maps have been deposited in EMD-50850. This mistake motivated us to inspect all of the maps and we found the same problem had occurred in the Type 3B maps. This was not noticed by the reviewer because we accidentally plotted the FSC curves from postprocessing from one refinement round before the one deposited in the EMD. We performed the same half-map shifting procedure for the Type 3B data and performed a final round of real-space refinement to produce new maps and models that have been deposited as EMD-50888 and 9FYP (superseding the previous entries).

**Reviewer #3 (Public Review):**
There are two minor points I recommend the authors to address:(1) In the response to Weakness 1, point (3), the authors state that "the Type 5 represented only 10-20% of the fibrils in the sample." However, this information is not labeled in the corresponding Figure 4. I suggest the authors verify and label all relevant percentages in the figures to prevent misunderstandings.

We aim to be as transparent as possible and this information was included in the main text however we did not label the percentage of Type 5 fibrils in Figure 4 because that would make the other percentages ambiguous. The percentages in Figure 4 represent the ratio of helical segments used for each type of refined structure in the dataset (always adding up to 100%), not the percent of all fibrils in the dataset. That is, there are sometimes untwisted or unidentifiable fibrils in datasets and these were not accounted for in the listed percentages. We have added a sentence to the Figure 4 legend to explain to what the percentages refer.

(2) While the authors have detailed the helical reconstruction procedure in the Methods section, it is necessary to indicate the scale bar or box size in the figure legend of the 2D representative classes to ensure clarity and reproducibility.

Thank you for reminding us to add the scale bars. This is now done for the 2D classes in Figures 11-17.

**Recommendations for the authors:**

**Reviewer #2 (Recommendations For The Authors):**
A critical look at the maps and models of the various structures at this stage may prevent the authors from entering suboptimal structures into the databases.

We agree. Thank you for suggesting this.

**Reviewer #3 (Recommendations For The Authors):**

The authors have responded adequately to these critiques in the revised version of the manuscript. There are two minor points.

(1) The authors state that "the Type 5 represented only 10-20% of the fibrils in the sample." However, this information is not labeled in the corresponding Figure 4. I suggest the authors verify and label all relevant percentages in the figures to prevent misunderstandings.(2) While the authors have detailed the helical reconstruction procedure in the Methods section, it is necessary to indicate the scale bar or box size in the figure legend of the 2D representative classes to ensure clarity and reproducibility.

Answered in public comments.